# LCRMP-1 is required for spermatogenesis and stabilises spermatid F-actin organization via the PI3K-Akt pathway

Jung-Hsuan Chang[1], Chia-Hua Chou[1], Jui-Ching Wu[1], Keng-Mao Liao[2], Wei-Jia Luo[1], Wei-Lun Hsu[1], Xuan-Ren Chen[3], Sung-Liang Yu [1], Szu-Hua Pan[2,3,4], Pan-Chyr Yang [5] & Kang-Yi Su [1,2,6 ✉]

Long-form collapsin response mediator protein-1 (LCRMP-1) belongs to the CRMP family which comprises brain-enriched proteins responsible for axon guidance. However, its role in spermatogenesis remains unclear. Here we find that LCRMP-1 is abundantly expressed in the testis. To characterize its physiological function, we generate LCRMP-1-deficient mice (*Lcrmp-1*$^{-/-}$). These mice exhibit aberrant spermiation with apoptotic spermatids, oligospermia, and accumulation of immature testicular cells, contributing to reduced fertility. In the seminiferous epithelial cycle, LCRMP-1 expression pattern varies in a stage-dependent manner. LCRMP-1 is highly expressed in spermatids during spermatogenesis and especially localized to the spermiation machinery during spermiation. Mechanistically, LCRMP-1 deficiency causes disorganized F-actin due to unbalanced signaling of F-actin dynamics through upregulated PI3K-Akt-mTOR signaling. In conclusion, LCRMP-1 maintains spermatogenesis homeostasis by modulating cytoskeleton remodeling for spermatozoa release.

[1] Department of Clinical Laboratory Sciences and Medical Biotechnology, College of Medicine, National Taiwan University, Taipei, Taiwan. [2] Genome and Systems Biology Degree Program, National Taiwan University and Academia Sinica, Taipei, Taiwan. [3] Graduate Institute of Medical Genomics and Proteomics, College of Medicine, National Taiwan University, Taipei, Taiwan. [4] Doctoral Degree Program of Translational Medicine, National Taiwan University, Taipei, Taiwan. [5] Department of Internal Medicine, College of Medicine, National Taiwan University, Taipei, Taiwan. [6] Department of Laboratory Medicine, National Taiwan University Hospital, Taipei, Taiwan. ✉email: suky@ntu.edu.tw

Globally, more than 30 million men suffer from infertility. The male infertility rate has often been underestimated because of ambiguous definitions of fertility and varying cultural beliefs, and it often lacks general awareness[1]. It is known that abnormal spermatogenesis causes male infertility. During spermatogenesis, the spermatogonia differentiate into spermatocytes, which undergo meiosis to become spermatids. Spermatids then undergo spermiogenesis to develop into mature spermatids. Spermiogenesis comprises acrosome formation, chromatin condensation, and flagellum development. The mature spermatids are then released from the Sertoli cells into the lumen of the seminiferous tubules in the process of spermiation[2–4]. Two actin-associated junctional cytoskeletal structures, ectoplasmic specializations (ESs) and tubulobulbar complexes (TBCs), appear near elongated spermatids in the apical Sertoli cell cytoplasm to accelerate the removal of the spermatid cytoplasm and the separation of the spermatids from the Sertoli cells. Abnormal spermiation can be attributed to the failure to start the process of spermiation, premature release of spermatids, defective removal of cytoplasm from spermatids, and unsuccessful disengagement, which leads to phagocytosis of the spermatids[5].

In the clinical setting, the causes of defective spermiation require a histological examination to identify and are difficult to diagnose. In addition, the specific molecular mechanisms behind spermiation are unclear. Successful spermiation depends on the phosphorylation and dephosphorylation of various molecules[5]. The process of spermiation is a crucial checkpoint in the production of sperm and thus strongly affects fertility. Previous studies have shown that F-actin and actin-associated cytoskeletal structure remodeling regulates spermatogenesis and spermiation, as in the case of apical ESs in spermiation and basal ESs in the blood–testis barrier (BTB)[6,7]. It is clear that the regulation of actin polymerization and stabilization is critical for spermatogenesis and spermiation. However, the role of some molecules, such as collapsin response mediator proteins (CRMPs), in regulating cytoskeletal structures involved in spermiation has been neglected.

The CRMP family has been well studied in terms of its role in controlling axon guidance and neurite outgrowth by regulating actin organization[8–10]. Long-form collapsin response mediator protein-1 (LCRMP-1) was identified as an isoform of CRMP-1 in the CRMP family[11]. CRMP-1 regulates neuron dendrite collapse and outgrowth in the brain, and cell migration generally, by controlling the dynamics of the actin cytoskeleton[12,13]. LCRMP-1 and CRMP-1 have opposing effects in the regulation of cancer invasion[11,14]. LCRMP-1 acts as an antagonist of CRMP-1, stabilizing F-actin and promoting the formation of filopodia in lung cancer metastasis. LCRMP-1 may have a function in spermiogenesis and spermiation by promoting actin polymerization and stabilization. A previous study found that 46 genes related to axon guidance were significantly upregulated under hormone suppression, which caused spermiation failure[15]. CRMP-4, another CRMP family member, is highly expressed in postmeiotic round spermatids and plays a role in spermiogenesis[16]. CRMP-1 is highly expressed in the brain and testes and is thought to be involved in spermatogenesis[17]. Further, clinical research has shown that CRMP-1 is downregulated in azoospermic males[18]. However, the physiological function of LCRMP-1 is still poorly understood. Its role in spermatogenesis via the regulation of actin cytoskeleton organization thus requires investigation.

Here, we generated mice deficient in LCRMP-1 ($Lcrmp$-$1^{-/-}$) to investigate its physiological functions. Although the $Lcrmp$-$1^{-/-}$ mice were viable and had no obvious abnormalities in the brain, they had reduced fertility due to low quantities of spermatozoa in the epididymis. Based on its relatively high expression level in the testes, we were able to identify the role of LCRMP-1 in

spermatogenesis. We discovered that it appears in a stage-dependent pattern in the seminiferous epithelial cycle and is specifically located near the spermatids in stages VII–VIII, indicating that its role is associated with the spermiation machinery. Further, LCRMP-1 deficiency caused F-actin disorganization near the lumen of the seminiferous tubules, leading to failed or delayed spermiation. This resulted in feedback upregulation of the PI3K–Akt–GSK3β/mTOR pathways and dysregulated actin dynamics during spermiation.

## Results

**LCRMP-1 is highly expressed in the testes**. Previous studies have shown that CRMP-1, as well as other CRMPs, is highly expressed in the brain. Similar to other CRMPs, the expression of CRMP-1 reaches its peak from the late embryonic stage to postnatal day 15[19,20]. However, the expression pattern of LCRMP-1 has not previously been well characterized. Our results showed that both CRMP-1 and LCRMP-1 are highly and ubiquitously expressed in the brain (Supplementary Fig. 1a). During embryogenesis, although the expression of LCRMP-1 is lower than that of CRMP-1, its pattern over time is comparable to that of CRMP-1 (Supplementary Fig. 1b). In addition to the brain, we found that among the other major organs, LCRMP-1 is most abundant in the testes in mRNA and protein levels (Fig. 1a, upper panel). In contrast with other organs, the expression level of LCRMP-1 in the testes, as indicated by the mRNA ratio of $Lcrmp$-$1$/$Crmp$-$1$ and protein expression, is higher than that of CRMP-1, suggesting its potential role in the male reproductive system (Fig. 1a, lower panel). To investigate its physiological function, we generated mice deficient in LCRMP-1 ($Lcrmp$-$1^{-/-}$) using a gene-targeting strategy (Fig. 1b). Mice deficient in LCRMP-1 were successfully identified via routine genotyping (Supplementary Fig. 1c). LCRMP-1 and CRMP-1 are spliced from exon 1a and exon 1b, respectively, using an identical C-terminus. For expression tracing, exon 1a was replaced by enhanced green fluorescent protein (eGFP). $Lcrmp$-$1$ was specifically deleted in a dose-dependent manner in wild-type ($Lcrmp$-$1^{+/+}$), heterozygous knockout ($Lcrmp$-$1^{+/-}$), and knockout ($Lcrmp$-$1^{-/-}$) mice, while CRMP-1 expression was intact, according to real-time quantitative polymerase chain reaction (qPCR) and Western blot assays (Fig. 1c, d). $Lcrmp$-$1^{-/-}$ mice were viable, suggesting that LCRMP-1 has no impact on embryogenesis. In addition, based on histopathological analysis, there were no obvious abnormalities in the major organs of these mice (Supplementary Fig. 2). However, there were more degenerating germ cells in the testis of $Lcrmp$-$1^{-/-}$ mice than in $Lcrmp$-$1^{+/+}$ mice by histopathological analysis (Supplementary Table 1). By applying the reverse genetic strategy, we compared $Lcrmp$-$1^{-/-}$ with $Lcrmp$-$1^{+/+}$ mice to investigate the function of LCRMP-1 in spermatogenesis.

***Lcrmp-1*-deficient mice exhibit reduced fertility and epididymal semen quality**. Based on the abundance of LCRMP-1 in the testes, we first assessed the fertility of $Lcrmp$-$1^{-/-}$ mice by mating them with female $Lcrmp$-$1^{+/+}$ mice. The fertility of male $Lcrmp$-$1^{-/-}$ mice was significantly reduced, as indicated by litter numbers, compared to the control (Table 1). In addition, 40% (two out of five) of the male $Lcrmp$-$1^{-/-}$ mice tested were infertile. In order to elucidate the cause of this reduced fertility, we further evaluated macroscopic and microscopic characteristics including quantity and quality of testes and sperm based on previous histopathological analysis. There were no apparent differences in the appearance, weight of the testes, sperm viability, and sperm morphology of $Lcrmp$-$1^{-/-}$ and $Lcrmp$-$1^{+/+}$ mice (Fig. 2a, b). However, there were substantially fewer spermatozoa in adult $Lcrmp$-$1^{-/-}$, even in mice aged 6–7 months, this substantial

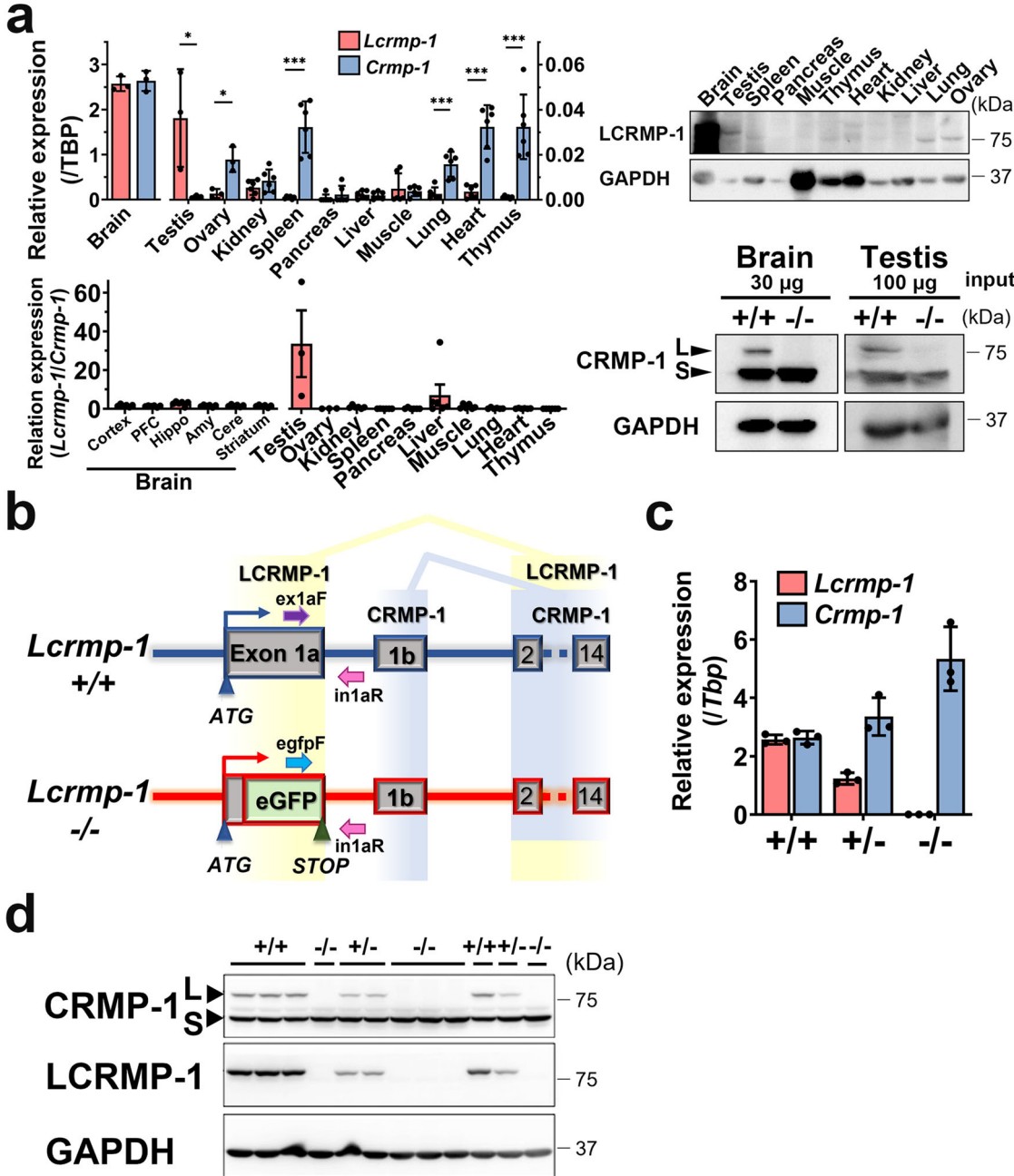

**Fig. 1 Expression profiling of LCRMP-1 and generation of LCRMP-1-deficient mice. a** Upper panel: real-time qPCR analysis of LCRMP-1 and CRMP-1 mRNA expression profile and Western blot analysis of LCRMP-1 protein expression profile by LCRMP-1 specific antibody in wild-type (*Lcrmp-1*$^{+/+}$) mice. Relative expression of qPCR was calculated as target gene/TBP. Lower panel: ratio of the relative mRNA expression (LCRMP-1/CRMP-1) (*n* = 6, mean ± SD, two-tailed Student's *t*-test, \**p* < 0.05, \*\**p* < 0.01, \*\*\**p* < 0.001) and Western blot analysis of LCRMP-1 and CRMP-1 protein expression by common CRMP-1 antibody. **b** The targeting strategy for *Lcrmp-1*. Exon 1a of the *Lcrmp-1*$^{+/+}$ allele was replaced with a target vector composed of E1a-eGFP (enhanced green fluorescence protein). Ex1aF and in1aR primer sets were used to detect the *Lcrmp-1*$^{+/+}$ allele, and egfpF and in1aR primer sets for the *Lcrmp-1*$^{-/-}$ allele. **c** Expression levels of LCRMP-1 and CRMP-1 mRNA in brain lysate from *Lcrmp-1*$^{+/+}$, *Lcrmp-1*$^{+/-}$, and *Lcrmp-1*$^{-/-}$ mice (*n* = 3, mean ± SD). **d** Expression levels of LCRMP-1 and CRMP-1 protein in brain lysate from *Lcrmp-1*$^{+/+}$, *Lcrmp-1*$^{+/-}$, and *Lcrmp-1*$^{-/-}$ mice. CRMP-1 has two forms: Long-form (L), is known as LCRMP-1, and short-form (S) is the same as CRMP-1. It had a dose-dependent effect on the expression of LCRMP-1. PFC prefrontal cortex, Hippo hippocampus, Amy amygdala, Cere cerebellum.

reduction was significant (Fig. 2c). We further tested the function of the spermatozoa. It was notable that the motility of the sperm in *Lcrmp-1*$^{-/-}$ was obviously lower (Supplementary Movie 1). By using the OpenCASA software analysis, the sperm from *Lcrmp-1*$^{-/-}$ exhibited fewer fast trajectories than *Lcrmp-1*$^{+/+}$ sperm (Fig. 2d). Except for motility, other velocity parameters including curvilinear velocity (VCL), straight-line velocity (VSL), and average-path velocity (VAP) were comparable to those in *Lcrmp-1*$^{+/+}$. In conclusion, mice deficient in LCRMP-1 exhibited a reduced number of spermatozoa. This led us to perform molecular histopathological analysis. Hematoxylin–eosin (H&E) staining revealed that the mean diameter of the seminiferous tubules in *Lcrmp-1*$^{-/-}$ mice was also significantly smaller than in *Lcrmp-1*$^{+/+}$ (154.93 ± 18.68 μm vs 186.51 ± 23.15 μm, *p* < 0.001)

| Table 1 Statistical analysis of fertility in *Lcrmp-1*[+/+] and *Lcrmp-1*[−/−] mice. | | | | | | | |
|---|---|---|---|---|---|---|---|
| **Male (*n* = 5)** | **Female (*n* = 5)** | **Litter number** | ***p*-value**[a] | **Pups** | ***p*-value**[a] | **Litter size**[b] **(mean ± SD)** | ***p*-value**[a] |
| +/+ | +/+ | 23 | 0.00894 | 165 | 0.0231 | 7.26 ± 0.80 | 0.0594 |
| −/− | +/+ | 10 | | 63 | | 3.72 ± 3.53 | |

[a]Two-tailed Student's t test.
[b]The litter size was calculated by dividing the number of pups by the number of litters.

(Fig. 2e). In summary, these results suggested that the low fertility of the *Lcrmp-1*[−/−] mice was possibly related to the reduced number and activity of spermatozoa, and indicated that LCRMP-1 deficiency may have impacts on the process after spermiogenesis.

**Spermatogonia and spermatocytes are more abundant in the seminiferous tubules of *Lcrmp-1*[−/−] mice.** We performed immunohistochemical analysis to test our hypothesis that LCRMP-1 deficiency affected late spermatogenesis. The targeting strategy we used enabled us to trace LCRMP-1 expression via immunoblotting with eGFP antibodies. First, our Western blotting analysis confirmed that the endogenous LCRMP-1 was replaced by eGFP in *Lcrmp-1*[−/−] mice, while *Lcrmp-1*[+/+] mice exhibited no eGFP signal (Fig. 3a). The eGFP signal was found to be most intense in the brain and testes among the major organs, which was consistent with the RNA expression profiles (Supplementary Fig. 3 and Fig. 3b). In the seminiferous tubules, the eGFP signal (representing the localization of LCRMP-1) was strong near the lumen suggesting that LCRMP-1 may be involved in the later stages of spermatogenesis (Fig. 3b). We further characterized subcellular localization of LCRMP-1 in the seminiferous tubules by co-staining with LCRMP-1, phalloidin, and DAPI (Fig. 3c). The result showed that LCRMP-1 was majorly cytoplasmic localization and partially colocalized with phalloidin. To test the staging of spermatogenesis in *Lcrmp-1*[−/−] mice, we identified the spermatogonia and spermatocytes via immunofluorescence staining with the specific antibodies synaptonemal complex protein 3 (SCP3) and DEAD-box helicase 4 (DDX4), respectively (Fig. 3d, e). The proportion of cells that expressed a positive signal during the early stages of spermatogenesis was significantly higher in *Lcrmp-1*[−/−] than in *Lcrmp-1*[+/+] mice. Further, the seminiferous tubule sections of all stages (I–XII) in *Lcrmp-1*[−/−] mice exhibited a notable increase in the abundance of early-stage germ cells(Supplementary Fig. 4). Taken together, these results suggest that the accumulation of early-stage germ cells may be caused by a defect in the late spermatogenesis.

**LCRMP-1 deficiency causes the accumulation of early-stage germ cells in the adluminal compartment of the seminiferous epithelium.** Based on previous studies, DDX4 (VASA) is expressed in early-stage germ cells such as spermatogonia, spermatocytes, and round spermatids, but is undetectable in elongated spermatids[21,22]. Since there were more DDX4-positive cells in *Lcrmp-1*[−/−] mice than in the *Lcrmp-1*[+/+], we further tested the distribution area and impact range of early-stage germ cells during spermatogenesis. During spermatogenesis, spermatogonia can differentiate into spermatocytes and spermatids as they move from the basal to the adluminal compartment. Based on DDX4 staining and nucleus morphology, we differentiated stages and classified specific cell types in different areas, such as spermatogonia, pachytene spermatocytes, round spermatids, and elongating spermatids, according to previous studies[22–24] (Fig. 4a). To precisely analyze and

quantify DDX4-positive early-stage germ cells among heterogeneous cell types in seminiferous tubules, we isolated the spermatids and spermatocytes via bovine serum albumin (BSA) density gradient sedimentation for fractionation followed by immunofluorescence staining. We obtained 28 fractions in total and classified into four groups: elongating spermatids fractions (E), round spermatids fractions (R), pachytene spermatocytes fractions (P), and others fractions (O). We found that LCRMP-1 was highly expressed in spermatids, while its expression was relatively lower in spermatocytes (Fig. 4b). The result indicated that DDX4-positive immature germ cells were significantly increased in ES and RS fractions of the *Lcrmp-1*[−/−] mice compared with those of *Lcrmp-1*[+/+] mice (Fig. 4c). Specifically, these results suggest that the absence of LCRMP-1 affected the late stages of spermatogenesis accompanied by the accumulation of early-stage germ cells.

**LCRMP-1 deficiency causes abnormal spermiation.** Our results suggested that LCRMP-1 may be involved in late spermatogenesis, when spermatids undergo spermiogenesis and spermiation. These stages are part of the transformation of spermatids to spermatozoa which are then released from the Sertoli cells[5]. In mice, the complete process of spermatogenesis takes about 35 days, which is four times longer than the period of a single seminiferous epithelial cycle from stage I to stage XII (8.6 days), and different stages of spermatogenesis occur in each segment of the seminiferous tubules. This pattern is called the "cycle of the seminiferous epithelium", which is divided into 12 stages (I–XII)[25–27]. The percentage of tubule cross sections that were in stages IV–VI was higher in *Lcrmp-1*[−/−] mice than in the wild type (Fig. 5a). Conversely, there were fewer tubules in stages VII–VIII. According to the morphology of each stage, stages VII–VIII in *Lcrmp-1*[−/−] mice exhibited excess cytoplasm surrounding the sperm tails (Fig. 5b, arrowhead). In addition, mature elongated spermatid heads were found after spermiation (stages IX–XII) in *Lcrmp-1*[−/−] mice (Fig. 5b, dotted circles). These results indicated that LCRMP-1 deficiency caused failed or delayed spermiation. We also analyzed the localization of LCRMP-1 during the cycle of the seminiferous epithelium. Interestingly, there were dynamic changes in the eGFP signal, which represented LCRMP-1 expression in seminiferous tubules of various stages (Fig. 5c). In late stage VIII, spermiation is completed and the preleptotene spermatocytes start transiting the BTB. LCRMP-1 was expressed in spermatocytes (especially primary and secondary) that were distributed in the adluminal area surrounded by the BTB area (Fig. 5c, stages VII–VIII). In stages IX–XII, the expression of LCRMP-1 gradually moved to the spermatids and residual bodies (Fig. 5c, stages IX–X and stages XI–XII). It was strongly expressed in the residual bodies during stages I–VI (Fig. 5c, stages I–III and stages IV–VI). In early stage VII, LCRMP-1 expression gradually diffused from the center of the lumen to the basement membrane site (Fig. 5c, stages VII–VIII). At the same time, it was also highly expressed along the dorsal curvature of the spermatid head (Fig. 5c, stages VII–VIII, arrowheads). It was obvious that the changes in

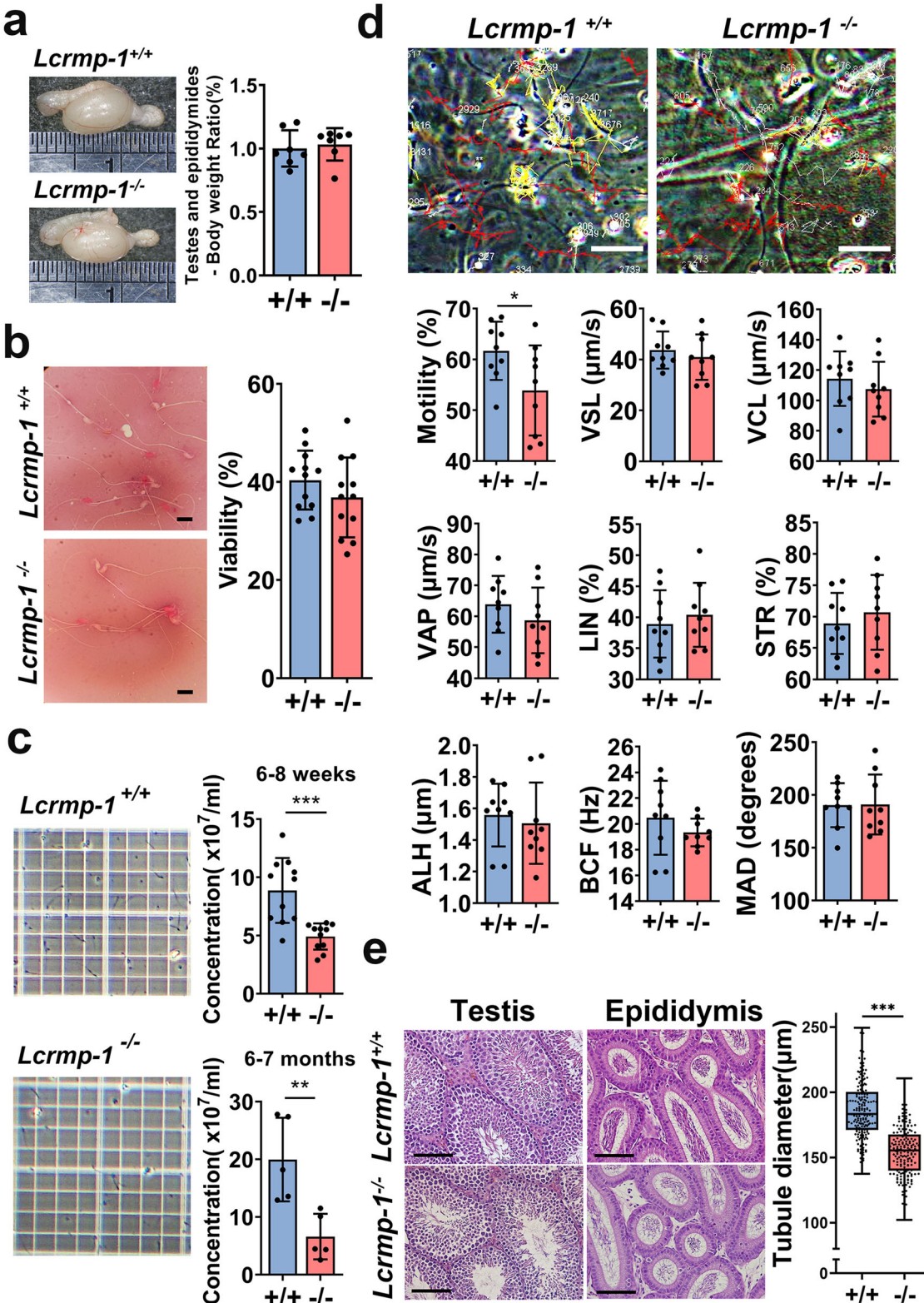

LCRMP-1 expression in the seminiferous tubules were cyclic and stage-dependent (Fig. 5c, and Supplementary Fig. 5a, b). Specifically, during stages VII–VIII, spermiation was defective and delayed in *Lcrmp-1*$^{-/-}$ mice relative to *Lcrmp-1*$^{+/+}$ mice (Fig. 5d). We observed that the elongated spermatid heads were localized near the Sertoli cell nuclei instead of undergoing disengagement near the lumen in stages VII–VIII (Fig. 5d, panels q, r, and s, arrowheads), and also observed large cells with pyknotic nuclei and reddish cytoplasm in the spermatid cell layer (Fig. 5d, panel k, arrowhead). Further, we observed degenerated spermatids or atypical residual bodies (Fig. 5d, panel t, arrowhead) and mature spermatids with excess cytoplasm (Fig. 5d, panels m, n, o, arrowheads). Taken together, these findings indicated that mice deficient in LCRMP-1 exhibit abnormalities in spermatogenesis, particularly the spermiation phase.

**Fig. 2 Lcrmp-1⁻/⁻ mice have fewer spermatozoa and reduced sperm motility. a** The size of testes and epididymides and the ratio of their weights to body weight ($n = 7$, mean ± SD). **b** The viability of sperm was analyzed using the eosin–nigrosin staining method. Viability was calculated as the number of white sperm cells divided by the total number of sperm cells (200 sperm cells for each mouse sample; $n = 12$, mean ± SD). Scale bars, 10 μm. **c** The concentration of spermatozoa in male mice of different ages (6–8 weeks and 6–7 months). The numbers were counted with a hemocytometer ($n = 11$ for 6–8 weeks, $n = 5$ for 6–7 months, mean ± SD, two-tailed Student's t-test, $**p < 0.01$, $***p < 0.001$). **d** Sperm motility measurement by OpenCASA software. Three 10 s videos for each mouse were recorded. The trajectories of the spermatozoa were analyzed. White indicates slow trajectories, yellow indicates moderate-speed trajectories, and red indicates fast trajectories. Scale bars, 50 μm. The following parameters of motility were also measured: motility rate, straight-line velocity (VSL), curvilinear velocity (VCL), average-path velocity (VAP), linearity (LIN), straightness (STR), amplitude of lateral head displacement (ALH), beat-cross frequency (BCF), and mean angular displacement (MAD). The original videos of the upper panel were showed in Supplementary movie 1 ($n = 9$, mean ± SD, two-tailed Student's t-test, $*p < 0.05$). **e** Left panel: H&E stain of the testes and epididymides of Lcrmp-1⁻/⁻ and Lcrmp-1⁺/⁺ mice. Right panel: diameter of the seminiferous tubules. Seminiferous tubules with round or nearly round were measured for each group. The diameters of 30 round tubules from each mouse were measured, and the mean diameters were calculated by averaging the long diameter and short diameter ($n = 6$, mean ± SD, two-tailed Student's t-test, $***p < 0.001$). Scale bars, 100 μm.

## LCRMP-1 deficiency leads to spermatid apoptosis as a result of spermiation abnormalities caused by F-actin disorganization.

LCRMP-1 has been reported to stabilize F-actin structures, and F-actin disorganization affects spermiation[14]. Therefore, we analyzed the organization of F-actin via phalloidin immuno-fluorescence staining followed by confocal microscope analysis. F-actin was located near the basal compartment, which harbored retained spermatids, in stages VII–VIII, and was less structured near the lumen of seminiferous tubules in stages IX–X, in Lcrmp-1⁻/⁻ mice than in Lcrmp-1⁺/⁺ mice (Fig. 6a). In addition, the F-actin exhibited a spoke-like pattern and a filamentous appearance in the tubules of Lcrmp-1⁻/⁻ mice (Fig. 6a, green arrowheads). To further characterize the organization of F-actin in Lcrmp-1⁻/⁻ mice, we performed immunofluorescence triple staining of SOX9 (Sertoli cell), phalloidin (F-actin), and DAPI followed by confocal microscope analysis (Fig. 6b). Our results revealed that F-actin was well-organized forming a nest-like or fence-like structure between the dorsal of spermatid head and the nuclei of the Sertoli cell that facilitated the removal of the residual bodies from spermatids in Lcrmp-1⁺/⁺ mice in stages VII–VIII while F-actin appeared wrapping around the heads and parts of tails of spermatid in Lcrmp-1⁻/⁻ (Fig. 6b, arrowheads). Moreover, in stages IX–X, the condensed and bundled F-actin appeared surrounding the tails of elongating spermatids in Lcrmp-1⁺/⁺ mice while the diffuse and disrupt F-actin organizations were observed in Lcrmp-1⁻/⁻ (Fig. 6b, arrowheads). These results indicated that LCRMP-1 deficiency caused disorganization of F-actin resulting in aberrant spermiation (Fig. 6c).

## The imbalanced Akt–mTOR–p70S6K/GSK3β axis interferes F-actin dynamics during spermatogenesis in mice deficient in LCRMP-1.

To further understand the mechanism by which LCRMP-1 mediates the process of spermiation, we focused on cytoskeletal arrangements in the late stages of spermatogenesis. Based on previous studies, we hypothesized that the F-actin dynamics and organization required for normal spermiation can be stabilized by either ribosomal protein S6 kinase (p70S6K) or LCRMP-1, regulated by protein kinase B (Akt)-downstream mammalian target of rapamycin (mTOR) and glycogen synthase kinase-3 beta (GSK3β), respectively[28,29]. Based on a previous finding that Akt is a key molecule regulating the ectoplasmic specialization dynamics in the seminiferous tubules at various stages[30], we initially tested the signaling involved in Akt-mediated F-actin dynamics (Fig. 7a). The expression of LCRMP-1 as control, compared with Lcrmp-1⁺/⁺ mice, phosphorylated Akt (p-Akt) and upstream phosphorylated phosphoinositide 3-kinase (p-PI3K) signaling were significantly upregulated in the testes of Lcrmp-1⁻/⁻ mice. The expression of downstream molecules, p-mTOR and p-GSK3β, was also enhanced as a result of Akt activation. Furthermore, we observed increased levels of cleaved caspase 3 in the testes of Lcrmp-1⁻/⁻ mice, suggesting that germ-

cell apoptosis may be the consequence of spermiation failure. Considering all the effects on heterogenous cell populations in the testis that are caused by LCRMP-1 deficiency, we separated testicular cells into high- and low-LCRMP-1-expressing fractions (HEF and LEF, respectively), based on LCRMP-1 and eGFP expression, for further validation (Fig. 7b). We found that p-Akt, p-mTOR, p-GSK3β, and p-p70S6K were upregulated in the LEF of Lcrmp-1⁻/⁻ mice, while cleaved caspase 3 was significantly upregulated in their HEF. Combined, these results suggest that LCRMP-1 deficiency causes F-actin disorganization during late spermatogenesis, followed by spermiation failure and germ-cell apoptosis. This deficiency may lead to an imbalance in two Akt-mediated signaling pathways, enhancing the mTOR–p70S6K signal and resulting in the F-actin disorganization phenotype in stages VII–X (Supplementary Fig. 6).

## Discussion

Male infertility and subfertility are commonly caused by abnormal spermatogenesis[31]. Although the genetic and environmental causes of abnormal spermatogenesis with reduced sperm quantity and function have been widely investigated[32–34], the molecular mechanism of spermiation, a crucial process that functions as a checkpoint determining the number of sperm produced, has been less well characterized than processes such as meiosis, mitosis, and spermiogenesis. This may be due to difficulties in defining the impact of spermiation failure on the number of released spermatids in the testes[35]. In the current study, we demonstrated that LCRMP-1 may play a role in consummating successful spermiation.

Specifically, Lcrmp-1⁻/⁻ mice have significantly reduced fertility, stemming from oligospermia. It has previously been reported that decreased sperm count is associated with spermiation defects[5,36]. Due to the unstable actin organization caused by LCRMP-1 deficiency, the process of spermiation was delayed and not completely successful, ultimately resulting in a lower number of spermatozoa being released. This led to excess cytoplasm surrounding the tail of sperm cells, potentially contributing to increased numbers of apoptotic spermatids in Lcrmp-1⁻/⁻ mice. In addition, the lack of LCRMP-1 caused increased DDX4 and SCP3 positivity in the seminiferous tubules. The presence of more DDX4-positive immature germ cells in the fractions where LCRMP-1 should have been highly expressed suggested that the accumulation of these cells overflowed into these fractions. Androgen suppression, which often causes abnormal spermatogenesis, contributes to the increased DDX4 expression associated with early meiotic prophase[37]. It is possible that numbers of meiotic and early-stage germ cells, which are DDX4 positive, accumulated due to abnormal spermiation in Lcrmp-1⁻/⁻ mice.

F-actin dynamics, including the formation of bundled and branched configurations, are important for spermiation[38–40]. The previous study demonstrated that LCRMP-1 was able to promote

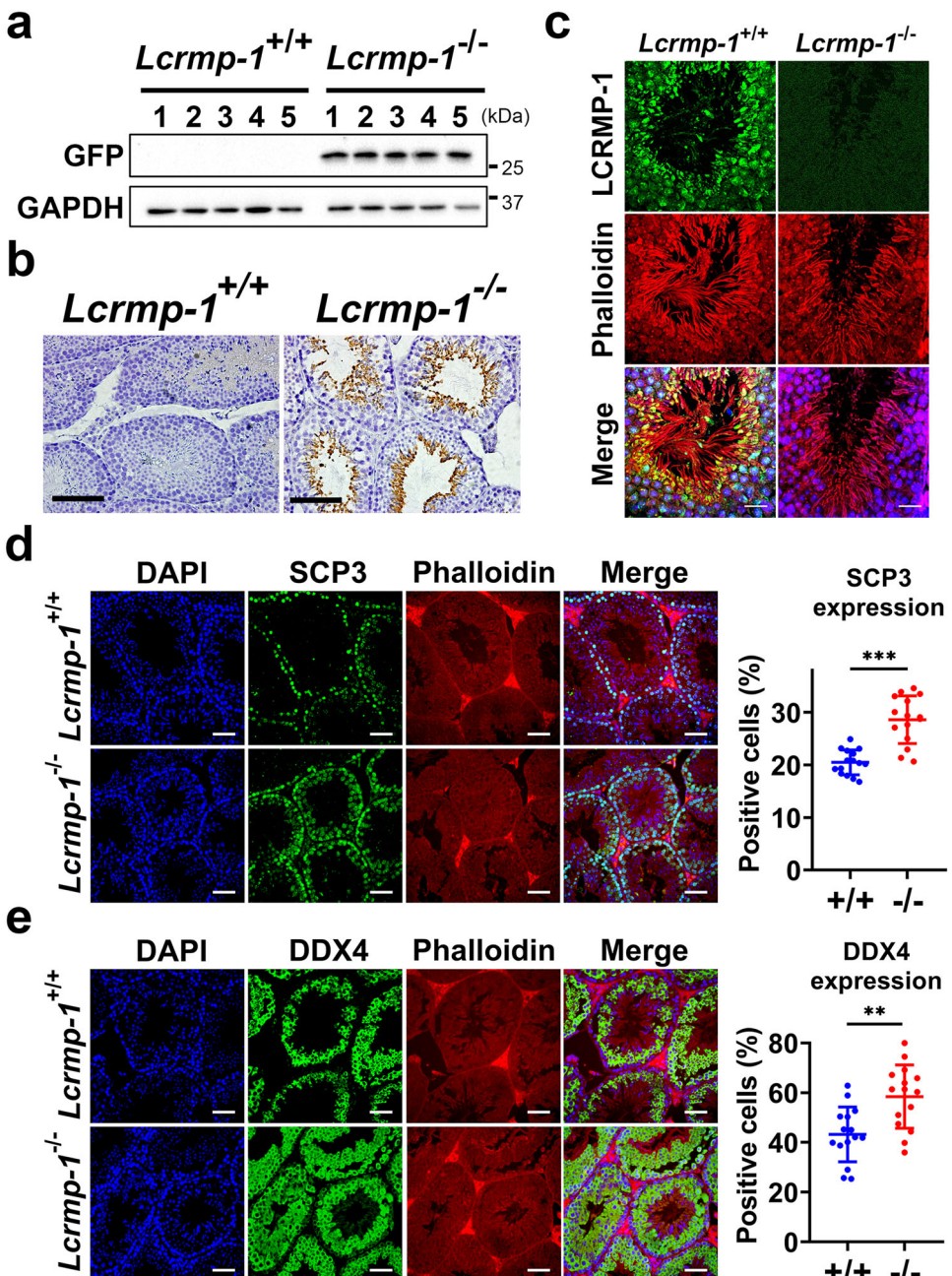

**Fig. 3 LCRMP-1 which is highly expressed near the lumen of seminiferous tubules affects immature germ cells. a** LCRMP-1 expression in the testes, as represented by eGFP, in a Western blotting analysis of *Lcrmp-1*⁺/⁺ and *Lcrmp-1*⁻/⁻ mice. The presence of the eGFP signal confirms that the *Lcrmp-1* gene locus had been replaced by the eGFP gene. **b** Immunohistochemical staining for LCRMP-1 expression patterns in the seminiferous tubules, as indicated by eGFP signaling. LCRMP-1 was highly expressed in the late spermatogenesis. Scale bars, 100 μm. **c** Immunofluorescence staining for LCRMP-1 localization in the seminiferous tubules using LCRMP-1 specific antibody (green), phalloidin-iFluor (red), and DAPI (blue) followed by confocal microscopy analysis. Scale bars, 20 μm. **d** SCP3 immunofluorescence staining of the testes of *Lcrmp-1*⁻/⁻ and *Lcrmp-1*⁺/⁺ mice. SCP3-positive cells were significantly more abundant in *Lcrmp-1*⁻/⁻ than in *Lcrmp-1*⁺/⁺ mice. The fluorescence signal was measured using ImageJ software and the ratio was calculated as the total number of Alexa Fluor 488 Dye (green) cells divided by the total number of DAPI (blue) cells (triplicate, *Lcrmp-1*⁺/⁺, $n = 5$; *Lcrmp-1*⁻/⁻, $n = 5$; mean ± SD, two-tailed Student's *t*-test, ***$p < 0.001$). Scale bars, 50 μm. **e** DDX4 immunofluorescence staining of the testes of *Lcrmp-1*⁻/⁻ and *Lcrmp-1*⁺/⁺ mice. DDX4-positive cells were obviously more abundant in *Lcrmp-1*⁻/⁻ than in *Lcrmp-1*⁺/⁺ mice. The fluorescence signal was measured using ImageJ software followed by the calculation of the ratio as the total number of Alexa Fluor 488 Dye (green) cells divided by the total number of DAPI (blue) cells (triplicate, *Lcrmp-1*⁺/⁺, $n = 5$; *Lcrmp-1*⁻/⁻, $n = 5$; mean ± SD, two-tailed Student's *t*-test, **$p < 0.01$). Scale bars, 50 μm.

cancer metastasis by regulating actin polymerization via the WAVE-1–Arp2/3 complex pathway[14]. It is possible that LCRMP-1 deficiency thus removes the function of stabilizing F-actin structures, leading to abnormal spermiation with unstable F-actin dynamics. In addition to being localized in the adluminal compartment of the seminiferous tubules, LCRMP-1 was highly expressed surrounding the spermatid head in stages VII–VIII. This indicates that LCRMP-1 is associated with the cytoskeletal structures in the spermiation machinery, which promote the release of spermatozoa from the Sertoli cells. The molecules

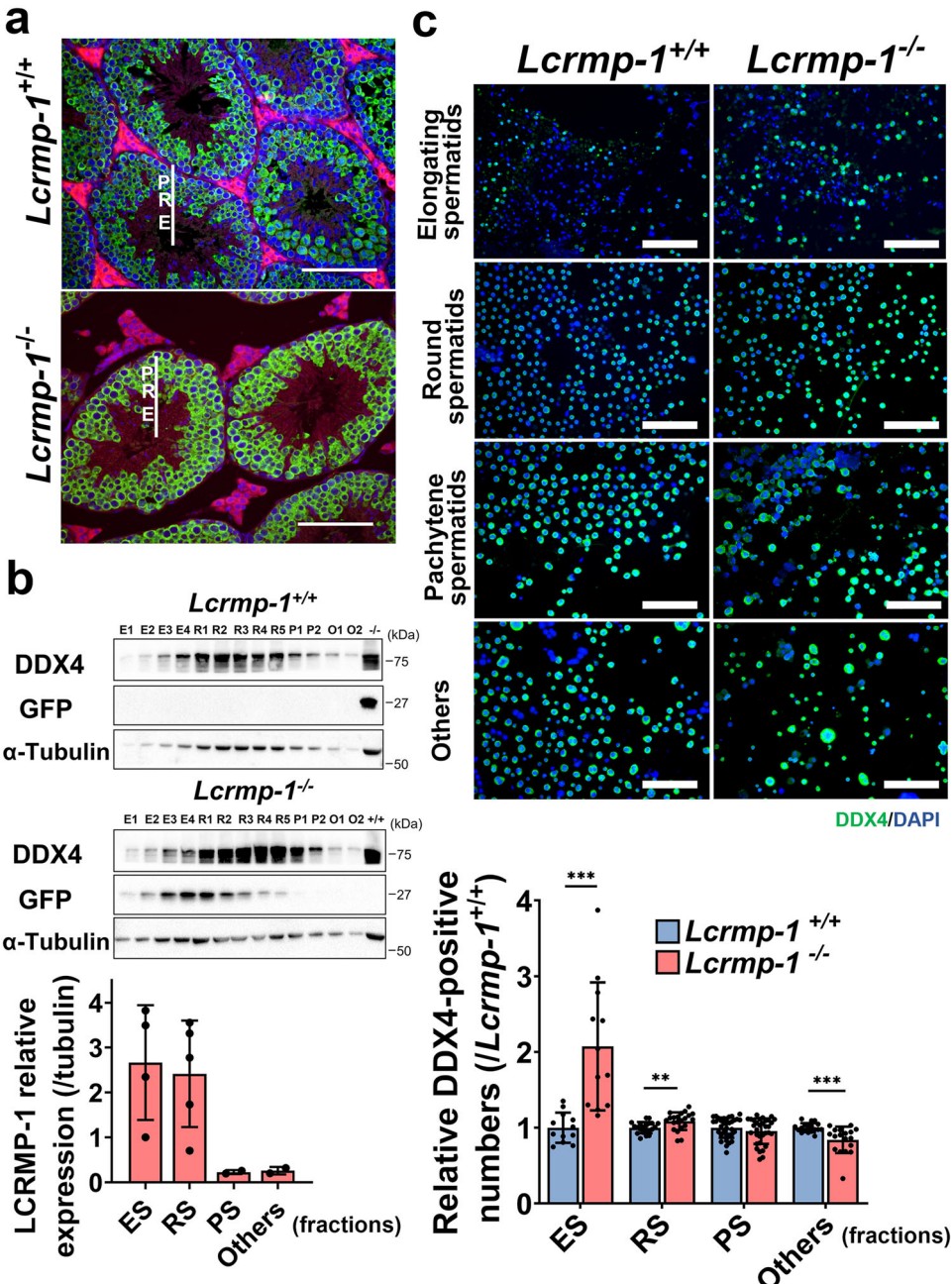

**Fig. 4 Deficiency of LCRMP-1 mainly expressed in spermatids causes DDX4-positive cells accumulation in the adluminal compartment. a** DDX4 expression pattern in the seminiferous tubules, as revealed by immunofluorescence staining. Blue, green, and red signals represent the nucleus, DDX4, and phalloidin (F-actin), respectively. P, pachytene spermatocytes; R, round spermatids, E, elongating spermatids. Scale bars, 100 μm. **b** Expression profiling of LCRMP-1 and DDX4 in fractionated testicular cells from seminiferous tubules. The testicular cells were isolated from the seminiferous tubules in the testes followed by BSA density-gradient fractionation. In total, 28 fractions were collected and 10% of the volume of each fraction was loaded for Western blotting. E1–E4, R1–R5, P1–P2, and O1–O2 represent corresponding elongating spermatids (fractions 1–4), round spermatids (fractions 5–11), pachytene spermatocytes (fractions 12–22), and others (fractions 23–28), respectively. We quantified GFP as an indicator to calculate the relative expression of LCRMP-1, normalized to E1. **c** Immunofluorescence staining of testicular cells from each cell fraction. Blue and green signals represent the nucleus and DDX4, respectively. The fluorescence positive signal ratio was measured using ImageJ. The ratio was calculated as the total number of Alexa Fluor 488 Dye-positive cells divided by the total number of DAPI-positive cells. Each cell fraction was analyzed in biological triplicate ($n = 3$, mean ± SD, two-tailed Student's $t$-test, **<0.01; ***$p < 0.001$). ES elongating spermatids, RS round spermatids, PS pachytene spermatocytes part. Scale bars, 100 μm.

involved in spermiation can be recycled within the Sertoli cells via cyclical molecule transport between the adluminal and basal compartments[35]. Interestingly, the expression pattern of LCRMP-1 in the seminiferous tubules was stage-dependent and cyclic, suggesting that it may be involved in recycling the spermiation machinery.

Mechanistically, our study indicated that a lack of LCRMP-1 leads to mTOR–p70S6K and PI3K–Akt–GSK3β phosphorylation. Both the PI3K–Akt–GSK3β and Akt–mTOR–p70S6K pathways have been reported to be involved in neurogenesis, spermatogenesis, and tumorigenesis[41–48]. GSK3β phosphorylates CRMP-2 to prevent it from inducing axon outgrowth in the

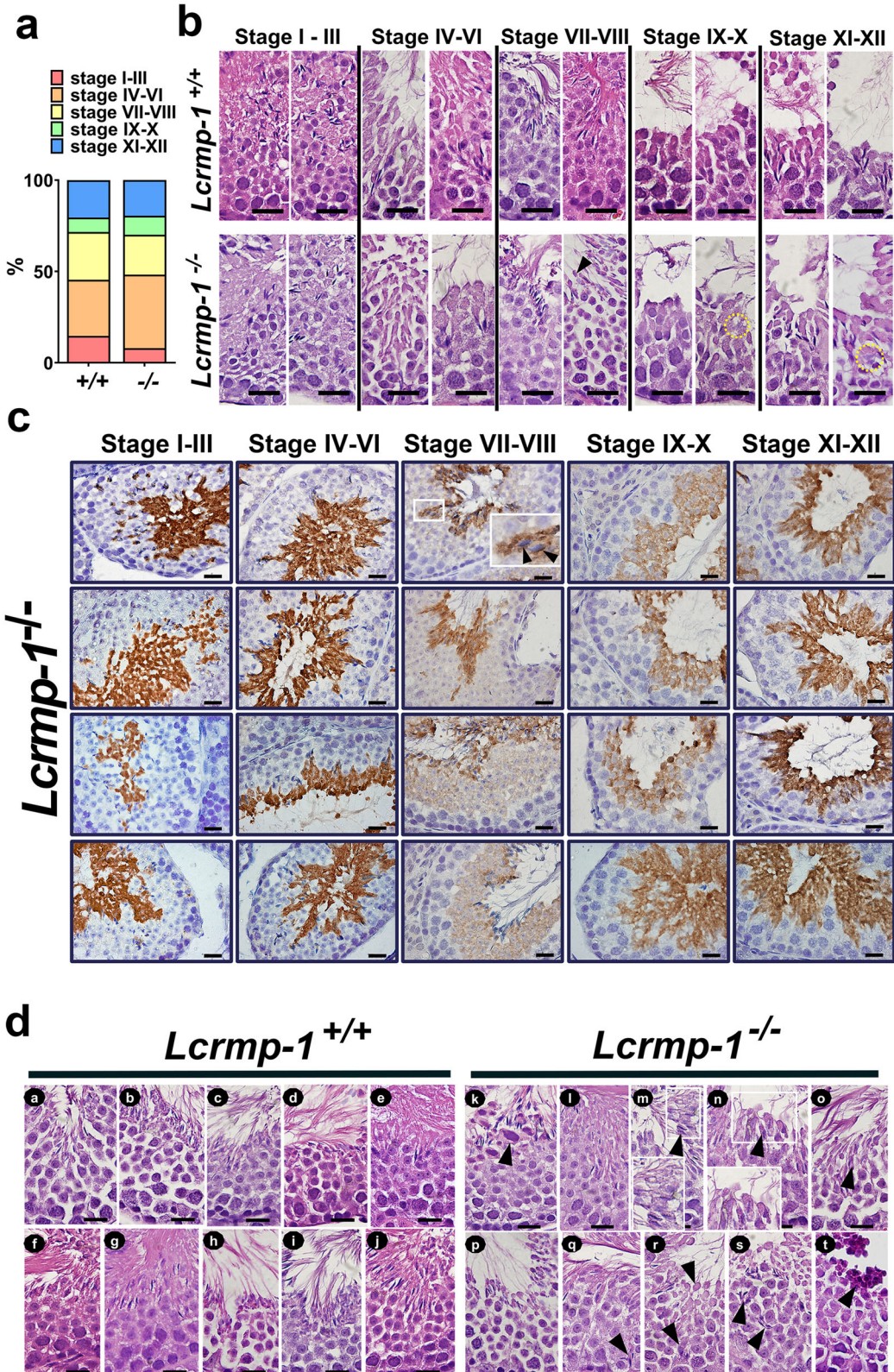

hippocampus[10], and it may induce the formation of filopodia by regulating the phosphorylation of LCRMP-1 at Thr-628[49]. In addition, the PI3K–Akt pathway controls GSK3β, which regulates F-actin remodeling and polymerization and triggers feedback regulations of its upstream pathway[48,50–52]. Taken together, PI3K–Akt–mTOR–p70S6K pathway is important in male reproduction, particularly in regulating the restructuring of apical ESs near the sperm head by controlling the actin cytoskeleton[28,30,53–56]. The previous study has shown that a local regulatory feedback mechanism between the adluminal and basal compartments controls early-stage cells and ES dynamics inducing the next cycle of spermatogenesis by indirect signaling of active peptides released from matrix metalloproteinase (MMP) proteolysis reaction during spermiation[57]. Therefore, in the case

**Fig. 5 LCRMP-1 is associated with spermiation and exhibits a stage-dependent pattern in the seminiferous tubule stages. a** Statistical analysis of the spermatogenesis stage. Testis sections were classified into 12 stages according to the histological morphology of cross sections of the seminiferous tubules. The 12 stages were further divided into five phases: I–III, IV–VI, VII–VIII, IX–X, and XI–XII. The percentages were calculated by dividing the number of seminiferous tubules in each phase by the total number of tubules analyzed (random fields, $Lcrmp-1^{-/-}$, $n = 5$, 87 tubules; $Lcrmp-1^{+/+}$, $n = 5$, 88 tubules). **b** Representative images of each phase. Sections of seminiferous tubules were prepared for H&E staining followed by morphological observation. The arrowhead in stages VII–VIII indicate residual bodies and excess cytoplasm, while the dotted circles in stages IX–XII indicate mature spermatid heads. Scale bars, 20 μm. **c** LCRMP-1 expression pattern during spermatogenesis in cycles of the seminiferous epithelium. Immunohistochemical staining of testis sections was done using eGFP antibodies to reveal the pattern of LCRMP-1 expression. The pattern in each stage was further analyzed. LCRMP-1 expression was cyclical, stage-specific, changing according to the stage of the seminiferous tubules. Black arrowheads in the white frame showing a magnified section indicate LCRMP-1 expression colocalized with cytoskeleton-like structures along the dorsal curvature of the spermatid head. Scale bars, 20 μm. **d** The histopathology of stage VII–VIII seminiferous tubules in $Lcrmp-1^{+/+}$ and $Lcrmp-1^{-/-}$ mice. The black arrowhead in (k) indicates a large cell with a pyknotic nucleus and reddish cytoplasm in the spermatid cell layer. Black arrowheads in (m), (n), (o), and (r) indicate spermatids with excess cytoplasm. Black arrowheads in (q), (r), and (s) indicate spermiation failure or delayed spermiation. The black arrowhead in (t) indicates degenerated spermatids or atypical residual bodies ($n = 5$ for $Lcrmp-1^{+/+}$ and $Lcrmp-1^{-/-}$ mice, 22 tubule cross sections showing stages VII–VIII). Scale bar, 20 μm.

of LCRMP-1 deficiency, spermatids in which LCRMP-1 should have been highly expressed, which are associated with the spermiation machinery, may have a feedback mechanism enabling them to upregulate the upstream molecules of LCRMP-1 in spermatocytes in which LCRMP-1 should have been weakly expressed. This is supported by a previous study showing that in spermatogenesis, actin organization regulates the Akt–mTOR pathway in a feedback loop[58]. Based on previous studies, CRMPs also served as either a cargo adapter transporting the Sra-1/WAVE1 and Arp2/3 complexes toward the adluminal compartments for supporting assembly and recycling of actin-based junctional structures or an actin-elongation factor binding protein facilitating actin polymerization and remodeling during spermiation[14,59–61]. Our results indicated that LCRMP-1 may promote the separation of sperm from the Sertoli cells by accelerating the conversion from the bundled actin microfilaments to branched actin filaments[38]. Conversely, the transportation of these complexes regulating actin assembly and stability becomes less efficient leading to reduced rates of actin polymerization and imbalanced F-actin/G-actin dynamics due to LCRMP-1 deficiency. In order to rescue the unstable F-actin organization and maintain the homeostasis of actin dynamics, the upregulated phosphorylation of PI3K–Akt–mTOR–p70S6K pathway compensates for this situation by inhibition of F-actin depolymerization through regulating cofilin[28,62,63]. Furthermore, based on previous studies, Ser9 phosphorylation of GSK3β can be an active form to phosphorylate Gli3 downregulating the Hedgehog pathway[64,65]. The Hedgehog pathway may be regulated by the actin cytoskeleton and PI3K–Akt–mTOR–p70S6K pathway, and modulate cytoskeleton reorganization in axon guidance, and contain a negative feedback loop[66–70]. From our perspective, without the interaction between GSK3β and LCRMP-1 in $Lcrmp-1^{-/-}$ mice, the level of phosphorylated GSK3β (Ser9) is increased, which may induce feedback loop to regulate the Hedgehog pathway promoting actin remodeling and affect the PI3K–Akt–mTOR–p70S6K pathway. However, this complicated network needs to be further investigated. In conclusion, LCRMP-1 deficiency exhibits a homeostatic imbalance of the two pathways leading to unstable F-actin dynamics and spermiation abnormalities.

There are some limitations to this study. First, it was difficult to rescue the infertility of $Lcrmp-1^{-/-}$ mice by re-expressing LCRMP-1. In addition, the difficulty of validating the role of LCRMP-1 in human infertility has not yet been completely overcome. Despite this, we identified an unclarified function of LCRMP-1 in the F-actin organization during spermatogenesis. Cytoskeleton organization is important not only for the anatomical architecture but also for normal physical functions. Recently, infections and environmental contaminants have been

shown to cause male infertility via dysregulation of cytoskeleton remodeling during spermatogenesis[71–73]. Further, male post-infection COVID-19 patients have been shown to have significantly reduced sperm concentration, although the sperm have normal morphology[74]. It would be worth investigating whether LCRMP-1-mediated spermiation machinery is involved in these conditions. In summary, we first identify a specific function of LCRMP-1 in spermiation and male fertility. In future studies, LCRMP-1-deficient mice can serve as a model for investigating spermiation and spermatogenesis, and LCRMP-1 may be studied for therapeutic or diagnostic applications.

## Methods

**Animals.** The LCRMP-1-deficient ($Lcrmp-1^{-/-}$) mouse was generated using a gene-targeting strategy based on the regular protocol. Briefly, exon 1a of LCRMP-1 was replaced in frame with enhanced green fluorescent protein (eGFP) in 129/sv mouse embryonic stem cells. The cells containing the targeted alleles were microinjected into blastocysts from C57BL/6 mice. These blastocysts were then implanted into a surrogate mother which gave birth to chimeric pups. These mice were mated with wild-type females, and the offspring were confirmed to be heterozygous (F1) mice via PCR genotyping. A syngeneic genetic background of $Lcrmp-1$ was achieved by backcrossing with the C57BL/6 mouse strain over 10 generations. We performed PCR with corresponding primers (Supplementary Table 2) to genotype LCRMP-1 homozygous knockout ($Lcrmp-1^{-/-}$), heterozygous knockout ($Lcrmp-1^{+/-}$), and wild-type ($Lcrmp-1^{+/+}$) mice. All animal experiments were approved by the Institutional Animal Care and Use Committee (IACUC) of National Taiwan University College of Medicine (IACUC number: 20201058). The mice were bred and maintained in pathogen-free micro-isolator cages with a standard light/dark cycle and free access to water and food.

**Animal fertility and sample collection.** For the fertility experiment, one male $Lcrmp-1^{+/+}$ or $Lcrmp-1^{-/-}$ mouse was paired with one female $Lcrmp-1^{+/+}$ mouse, and this was repeated five times per genotype, yielding five replicates. The male mice were at least two months old and the females were one month old at the start of the experiment, and they were housed together for four months. The numbers of pups and litters produced in each cage were recorded for calculating fertility.

For sample processing, the major organs of 6–8-week-old male mice were collected for analysis. To collect embryos, pregnant mice were sacrificed and the embryos were collected from the uterus under the dissecting microscope. For the early embryonic stages (embryonic day 6.5 to 11.5 [E6.5–E11.5]), whole embryos were collected, and for the later stages (E12–E18) the brains of the embryos were isolated for further analysis. When collecting the major organs, the organs were isolated from adult mice and then subjected to either 4% formaldehyde fixation for tissue sectioning or RNA and protein extraction for analysis.

**RNA extraction and quantitative analysis.** Mouse organs were lysed in 1 ml TRI reagent (Sigma-Aldrich, USA) and homogenized with TissueLyser II homogenizer (QIAGEN, Germany). We added 200 μl chloroform to each sample and transferred the upper aqueous layer to a tube containing 500 μl isopropanol. After centrifugation, the supernatant was removed and 500 μl 75% ethanol was added for washing. We air-dried the pellet, then resuspended it in diethylpyrocarbonate (DEPC)-treated water. For reverse transcription, we used High Capacity cDNA Transcriptase Kits (Thermo Fisher Scientific, USA) according to the user's manual. Briefly, we mixed 2 μg total RNA with 10× reverse transcriptase (RT) buffer, 25× deoxynucleotide triphosphates (dNTP), 10× RT random primers, and RT, and performed RT-PCR under the following conditions: 25 °C, 10 min; 37 °C, 120 min; 85 °C, 5 min; 4 °C. We added 30 ng of cDNA products with 10× SYBR green master mix and primers to

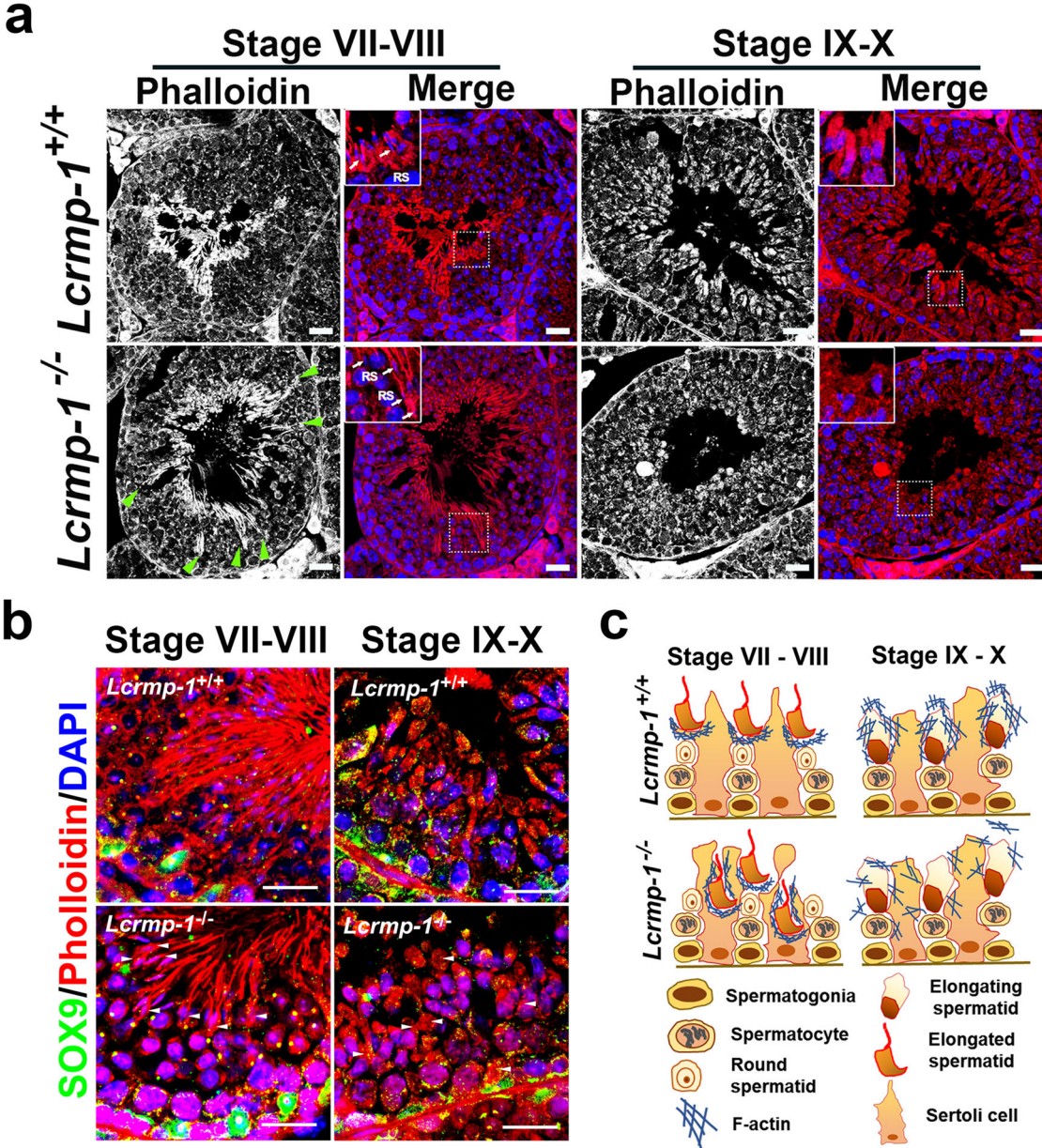

**Fig. 6 Disorganized F-actin in *Lcrmp-1*⁻/⁻ mice. a** Testis sections were stained using DAPI and phalloidin and subjected to confocal microscope analysis. Green arrowheads indicate retained spermatids near the basal compartment, while the white arrows in the white frames showing magnified sections indicate mature elongated spermatids at stages VII–VIII. In stages XI–X, the white frames show magnified images of the elongating spermatids near the lumen of the seminiferous tubules. The phalloidin results were shown as grayscale images. Scale bars, 20 μm. **b** Immunofluorescence staining of SOX9 (Sertoli cell), phalloidin (F-actin), and DAPI in stages VII–VIII and IX–X of *Lcrmp-1*⁺/⁺ and *Lcrmp-1*⁻/⁻ mice seminiferous tubules followed by confocal microscope analysis. Arrowheads in stages VII–VIII indicate F-actin wrapping around mature spermatids, and arrowheads in IX–X show disorganized F-actin in elongating spermatids. Scale bars, 20 μm. **c** Illustration of disorganized F-actin in *Lcrmp-1*⁻/⁻ mice in comparison to F-actin in wild-type mice. The filamentous F-actin retains mature spermatids near the basal compartment of the seminiferous epithelium and surrounds the tails and heads of spermatids during separation from the Sertoli cells in stages VII–VIII as shown in (**a**), (**b**). The disorganized F-actin appears near the tails of elongating spermatids in stages IX–X in *Lcrmp-1*⁻/⁻ mice as shown in (**a**), (**b**).

perform a qPCR analysis, using the Applied Biosystems QuantStudio 3 Real-Time PCR System (Thermo Fisher Scientific, USA), under the following conditions: 50 °C, 2 min; 95 °C, 10 min; 40 cycles of 95 °C for 15 s and 60 °C for 1 min. The messenger RNA (mRNA) expression level was normalized to the gene expression level of TATA-binding protein (TBP). We performed three technical replicates for each sample and calculated the values using the comparative threshold cycle method ($2^{-\Delta Ct}$). The qPCR primers are listed in Supplementary Table 2.

**Protein extraction and Western blotting**. Testes and brain tissues were lysed in T-PER Tissue Protein Extraction Reagent (Cat. #78510, Thermo Fisher Scientific, USA) with Phosphatase Inhibitor Cocktail (Cat. #4906837001, Roche, USA) and Protease inhibitor cocktail (Cat. #S8830, Sigma-Aldrich, USA). After the tissues had

been homogenized in a TissueLyser II (QIAGEN, Germany), the lysates were centrifuged at 14,000 rpm for 20 min and the supernatant was sonicated. We then loaded 30 μg (whole-tissue lysate), 13 μg (fractionated testicular cells), and 200 μg (CRMP-1 and LCRMP-1 expression in testis) protein with 4× sample buffer into each well. The samples were electrophoresed on 12% or 10% sodium dodecyl sulfate-polyacrylamide gels and transferred onto polyvinylidene difluoride membranes (Millipore, USA). These membranes were blocked with 1% nonfat milk or 1% BSA and incubated with primary antibodies overnight at 4 °C (anti-CRMP-1, Cat. #sc-365348, Santa Cruz biotechnology, 1:1000; anti-CRMP-1, Cat. #PA5-34768, Thermo Fisher Scientific, 1:1500; anti-LCRMP-1, homemade[14], 1:10,000; anti-GSK3β, Cat. #9315, Cell Signaling Technology, 1:1000; anti-phosphorylated GSK3β, Cat. #9336, Cell Signaling Technology, 1:1000; anti-GFP, Cat. #2956, Cell Signaling Technology, 1:1000; anti-cleaved caspase

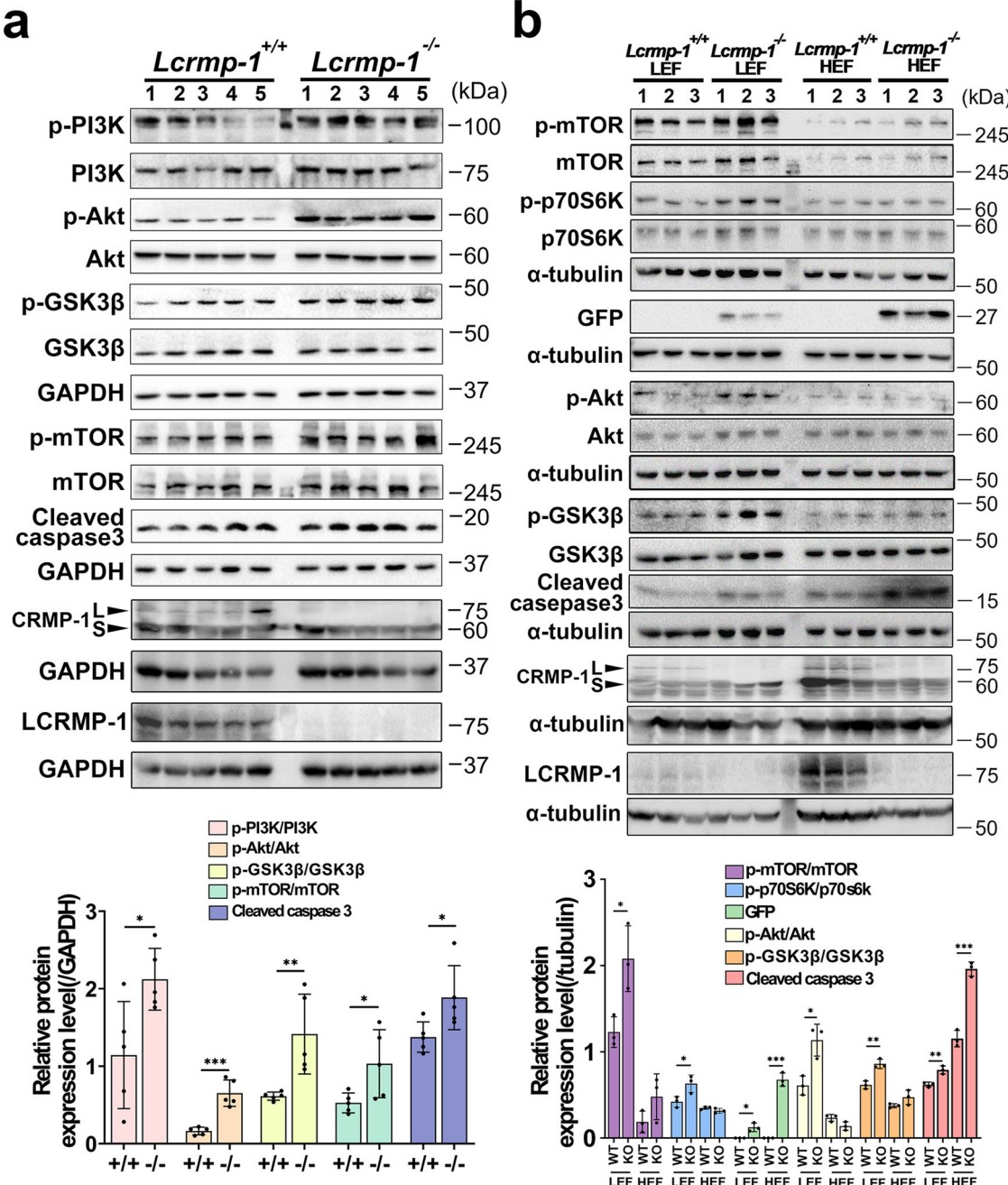

**Fig. 7 Highly phosphorylated upstream molecules of LCRMP-1 in _Lcrmp-1⁻/⁻_ mice. a** Western blotting analysis and quantification of whole-testis lysate for PI3K–Akt–mTOR/GSK3β signaling related to F-actin dynamics in _Lcrmp-1⁺/⁺_ and _Lcrmp-1⁻/⁻_ mice ($n = 5$, mean ± SD, two-tailed Student's _t_-test, *$p < 0.05$, **$p < 0.01$, ***$p < 0.001$). **b** Western blotting analysis and quantification of fractionated testicular cells (spermatids, spermatocytes, and others) of _Lcrmp-1⁺/⁺_ and _Lcrmp-1⁻/⁻_ mice. HEF indicates pooled cell fractions 1–8, which exhibit high expression of LCRMP-1, while LEF indicates pooled cell fractions 9–20, which exhibit low expression of LCRMP-1 ($n = 3$, mean ± SD, two-tailed Student's _t_-test, *$p < 0.05$, **$p < 0.01$, ***$p < 0.001$).

3, Cat. #9661, Cell Signaling Technology, 1:1000; anti-mTOR, Cat. #2972, Cell Signaling Technology, 1:1000; anti-phosphorylated mTOR, Cat. #2971, Cell Signaling Technology, 1:1000; anti-Akt, Cat. #9272, Cell Signaling Technology, 1:1000; anti-phosphorylated Akt, Cat. #9271, Cell Signaling Technology, 1:1000; anti-PI3K, Cat. #4292, Cell Signaling Technology, 1:1000; anti-phosphorylated PI3K, Cat. #4228, Cell Signaling Technology, 1:1000; anti-p70S6K, Cat. #2708, Cell Signaling Technology, 1:1000; anti-phosphorylated p70S6K, Cat. #9234, Cell Signaling Technology, 1:1000; anti-α tubulin, Cat. #11224-1-AP, proteintech, 1:20,000; anti-GAPDH, Cat. #60004-1-Ig, proteintech, 1:20,000; anti-DDX4, Cat. #ab13840, Abcam, 1:2000) followed by incubation with horseradish peroxidase (HRP)-conjugated antibody for an hour at room temperature. We used enhanced chemiluminescent substrates (ECL) (Millipore, USA) and recorded the chemiluminescent signal using ChemiDoc XRS+ (Bio-Rad, USA). We used ImageJ software for quantification.

**Sperm analysis.** After the mice had been sacrificed, we harvested the epididymides and testes immediately. The epididymis was dissected and spermatozoa were released in Krebs–Ringer medium[75]. The spermatozoa suspension was collected for counting using the counting chamber. To analyze viability, the spermatozoa were collected for eosin–nigrosin staining (1% eosin and 10% nigrosin in normal saline, [2:1] by volume, Sigma-Aldrich, USA). The spermatozoa suspension was incubated for 15 min in two steps (swim-up and swim-out steps) at 37 °C. To evaluate the motility of the sperm, high-motility spermatozoa were collected from the upper-most 90% (by volume) of the spermatozoa suspension. We recorded videos of the spermatozoa suspensions for OpenCASA analysis, which assessed motility, VSL, VCL, VAP, linearity (LIN), straightness (STR), amplitude of lateral head displacement (ALH), beat-cross frequency (BCF), and mean angular displacement (MAD)[76].

**Immunohistochemical staining, H&E staining, and histopathological analysis**. We performed immunohistochemical staining according to the standard procedure. Briefly, the 5μm tissue slides were twice deparaffinized in a xylene bath for 5 min. The slides were then sequentially rehydrated in 100%, 90%, 80%, 70%, and 50% ethanol and retrieved in citrate buffer (pH 6.0) in a Bio SB TintoRetriever Pressure Cooker for 15 min. The tissues were quenched with 3% $H_2O_2$ and incubated with primary antibody (anti-GFP, Cell Signaling Technology, 1:200) diluted with Antibody Dilution Buffer (Ventana, Roche, USA) overnight at 4 °C. The slides were then incubated in N-Histofine Simple Stain Mouse MAX PO (M) or (R) secondary antibody (Nichirei Biosciences Inc., Japan) at room temperature for 30 min. We stained the tissues using the Dako Liquid DAB + Substrate Chromogen System (Agilent, USA) and hematoxylin (Sigma-Aldrich, USA). The slides were then mounted using Micromount Mounting Medium (Cat. #3801731, Leica, Germany) and images were acquired using an Olympus DP70. The H&E staining and histopathological interpretation of mouse organs were performed by the pathologist from the Taiwan Mouse Clinic and Laboratory Animal Center, College of Medicine, National Taiwan University Medical College. Each organ section was examined at lower-power field and quantified with a severity grading scheme (Grading scheme I) based on the previous study[77]. The diameter of the seminiferous tubule calculation was according to the previous study[78]. Briefly, only the round or nearly round cross sections of the seminiferous tubules were chosen randomly and measured for each group. The diameter of the seminiferous tubule was measured and averaged across the long and short axes. Totally 180 round tubules from 6 mice in each group were obtained for analysis.

**Immunofluorescence staining**. For immunofluorescence staining, slides were deparaffinized and retrieved in citrate buffer (pH 6.0) or Tris-EDTA (pH 9.0) at 114–121 °C for 15 min and incubated in Rodent Block M (Biocare Medical, USA) for an hour at room temperature. The slides were incubated in primary antibody (anti-SCP3, Cat. #ab97672, Abcam, 1:100; anti-DDX4, Cat. #8761, Cell Signaling Technology, 1:100, Cat. #ab13840, Abcam, 1:100; anti-LCRMP-1, homemade, 1:100; anti-SOX9, Cat. #H00006662-M01, Abnova, 1:100) diluted with Antibody Dilution Buffer (Ventana, Roche, USA) or 1% BSA phosphate-buffered saline with Tween 20 (PBST) overnight at 4 °C and then in secondary antibody conjugated to Alexa Flour 488 (Cat. #A-11001, Cat. #A-11008, Thermo Fisher Scientific, 1:500) for an hour at room temperature. The samples were stained with Phalloidin (Cat. #ab176757, Abcam, UK) and Hoechst (DAPI) (Cat. #33342, Thermo Fisher Scientific, USA) for an hour and mounted using Fluoromount Aqueous Mounting Medium (Cat. #F4680, Sigma-Aldrich, USA). The results were quantified using ImageJ software. Olympus IX83 and Olympus DP70 were used to capture the images, and the confocal microscopy images were obtained with the Zeiss LSM 880 and Zeiss LSM 780.

**Enrichment of testicular cells**. To extract and enrich the testicular cells, we dissected the mouse testes and removed the tunica albuginea. We used a procedure based on BSA density isolation[79]. In brief, the testes were incubated in Krebs buffer with type IV collagenase (Cat. #C5138, Sigma-Aldrich, USA) for 3 min at 37 °C with gentle agitation. After digestion of connective tissue, the released tubules were washed and incubated with trypsin solution (trypsin, Cat. #T1426, Sigma-Aldrich, USA; DNase I, Cat. #10104159001, Sigma-Aldrich, USA) for 15 min at 34 °C at 15 rpm. The testicular cells were filtered through a 40 μm cell strainer on ice. After centrifuging twice at $600 \times g$ for 5 min at 4 °C, the cells were filtered again and loaded into a BSA-gradient solution (5%, 4%, 3%, 2%, 1% BSA from the bottom to the top of the solution). The cells underwent gravity sedimentation for 1.5 h, and 28 fractions were then collected by collecting 1 ml from each layer from the top to the bottom. The cells were centrifuged and each fraction was washed. The cell pellets were then frozen for Western blotting or fixed with 4% paraformaldehyde (PFA) for immunofluorescence staining.

**Statistics and reproducibility**. The sample size was not predetermined but based on previously conducted pilot tests. Based on the number of replicates required to obtain consistent results, the experiments were repeated using independent mouse replicates (biological replicates). All experiments were analyzed using two-tailed Student's t-tests conducted in Graphpad Prism 8. P-values of <0.05 were considered statistically significant. Grouped data are presented as mean ± standard deviation (SD).

**Reporting summary**. Further information on research design is available in the Nature Portfolio Reporting Summary linked to this article.

## Data availability
All data generated or analyzed during this study are included in this article and the supplementary Information. The raw data for Figs. 1–5 and Fig. 7 are available in Supplementary Data 1. The original Western blotting images are available in Supplementary Fig. 7.

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

## Acknowledgements

We thank Prof. Pei-Shiue Tsai and Ms. Tse-En Wang from School of Veterinary Medicine National Taiwan University for providing technical assistance with sperm analysis. We thank the imaging core at the First Core Labs, National Taiwan University College of Medicine, for the technical support. We also thank National Taiwan University Laboratory Animal Center for the histopathological examination and technical assistance. This work was supported by grants MOST105-2628-B-002-051-MY3 (K.Y.S.), MOST110-2314-B-002-269 (K.Y.S.), and MOST111-2628-B-002-029-MY3 (K.Y.S.) from Ministry of Science and Technology, Taiwan.

## Author contributions

J.-H.C. performed all the experiments and analyses, and C.-H.C. provided the experimental assistance. K.-Y.S. generated the *Lcrmp-1*$^{-/-}$ mouse model. J.-H.C., K.-M.L., and W.-J.L. performed mice breeding. J.-H.C. and K.-Y.S. designed this study. S.-L.Y., P.-C.Y., and K.-Y.S. supervised the research. J.-C.W., W.-L.H., X.-R.C., S.-L.Y., and S.-H.P. provided the technical assistance, reagents, and equipment. J.-H.C. and K.-Y.S. wrote and revised this manuscript. All the authors read and approved the final manuscript.

## Competing interests

The authors declare no competing interests.
