## [Peer Review File · Communications Biology]

Reviewers' comments:

Reviewer #1 (Remarks to the Author):

This study examined the physiological role of LCRMP-1, which showed strongest expression in the testis, by producing knockout mice and studied the phenotype of them closely. The KO mice showed normal development and growth but the sperm production was significantly reduced. This reduction was then revealed caused by abnormal spermiation. As the LCRMP-1 was supposed to take roles in actin dynamics, authors examined the signaling cascade regarding to F-actin organization to show upregulation of some relating molecules.

This study was performed with sophisticated manner resulting clear and solid data, supporting the conclusion of this manuscript.

I have noticed some minor concerns in the manuscript as described below.

Line 108, (Supplementary Table 1); In the supplementary Table 1, I'm not sure that each number in the columns represents.

In Supplementary Table 1, it is not clear what each number in the column represents. What pathological scores actually represent is also not clear. The lesions in an organ would be completely different from those in other organs. The score 5 = severe/high is added with 76-100%, which is also confusing.

Line 120, Fig. 2b; The tubular diameter was measured to be around 500 μm in this experiment. It is strange however that regular mouse seminiferous tubules are having about 200 μm in diameter. I suspect that authors used incorrect measure or miscalculated.

Line 122 – 135; The story line in this part is irregular and a bit confusing. A simple conclusion seems to be "reduced number of spermatozoa in KO mice" in the sample collected from epididymis. Reduced motility is another abnormal data. Other than those, spermatozoa in epididymis were similar to those in control. Such fact can be written in a simple way.

Line 149, Fig. 3c; The staining of SCP3 seems not specific to spermatocytes, but even the interstitial tissue stained also. I think the antibody is not good enough. At closer look, the DDX4 antibody was not so specific to germ cells either.

Line 154;

Here, defects in late spermatogenesis were suspected to be a cause of accumulation of early-stage germ cells. Is it correct to interpret that the decrease in late-stage germ cells created an open space for early-stage germ cells to accumulate? Are there any other possible mechanisms?

Line 171 -175; Relating to the above comment, the Fig. 4c showed that number of ES was higher in KO mice than control, while RS and PS were almost same in both groups. Doesn't this result contradict with data in Fig. 3, which insisted increased number of early-stage germ cells, including PS, in KO mice?

Line 216; "nearer" may be near.

Reviewer #2 (Remarks to the Author):

Comments for "LCRMP-1 consummates spermiation in male reproduction by stabilizing F-actin organization via PI3K-Akt pathway balancing".

Chang J-H et al. examined the role of long-form collapsing response mediator protein-1 (LCRMP-1) in the spermatogenesis based on their findings including abundant expression of LCRMP-1 in testis and reduced fertility of LCRMP1-1 knockout (KO) mice. They found that 1) lower motility of LCRMP-1 KO sperm; 2) involvement of LCRMP-1 in the late stages of spermatogenesis; 3) hyperactivation of Akt-mTOR signaling in LCRMP-1 KO testes. They propose LCRMP-1 deficiency causes imbalanced Akt-downstream signaling in turn to bring disorganized F-actin assembly in late spermatogenesis, followed by spermiation failure and germ-cell apoptosis. These findings are interesting and may clarify the molecular mechanism of LCRMP-1-regulated F-actin cytoskeleton in the spermatogenesis. However, this manuscript is missing several data supporting the role of LCRMP-1 in spermatogenesis including LCRMP-1 protein (not eGFP) expression and subcellular localization and the direct relation of LCRMP-1 and F-actin in spermatids. The molecular machinery of compensatory upregulation of Akt-signaling in the absence of LCRMP-1 is not discussed.

- 1) Selective expression of LCRMP-1 in testis is shown by mRNA expression (Fig. 1a). This should be confirmed by protein expression of LCRMP-1 and CRMP1 in testes from wild-type and knockout animals by immunoblot analysis.
- 2) Genotypes of Fig. 5c panels are missing. Multiple panels of the same stages (4 panels of VII-VIII; 2 panels of IX-X; 2 panels of XI-XII) should be explained.
- 3) The expression of LCRMP-1 in seminiferous tubules is shown by eGFP immunohistochemistry (IHC) of LCRMP-1 KO in Fig. 3b. This may be sufficient to show stage-specific LCRMP-1 expression, however, eGFP signal cannot represent the subcellular localization of LCRMP-1. Subcellular localization of LCRMP-1 in developing spermatids along with F-actin-staining should be presented with higher magnification.
- 4) It is difficult to image the illustrated F-actin distribution (Fig. 6a right panels) from the Phalloidin staining (Fig. 6a left and middle panels). Higher magnification view is required (inset pictures are insufficient). Triple immunostaining with Sertoli cell marker may be examined to show the different distribution of F-actin in LCRMP-1 KO seminiferous tubules.
- 5) Fig. 6b and 6c panels should include LCRMP-1 and CRMP1 immunoblots.
- 6) Several scenarios for the molecular machinery of compensatory upregulation of Akt-signaling in the absence of LCRMP-1 may be proposed in the discussion.
- 7) L237, eGFR should be eGFP

Response summary

The following letter contains full point-by-point responses to both reviewers. The first reviewer had several comments related to spermatogenesis and the second reviewer had several comments related to neuroscience especially CRMPs family proteins (7 from the reviewer #1 and 7 from the reviewer #2). We briefly summarized responses in below with a hyperlink to each detail description within the letter.

For the convenience of reviewers to read, we additionally provided a clear manuscript file with line number labeled as the supplemental information. The line number in the response letter was based on this supplementary file.

Reviewer #1

Comment 1: Unclear description of Supplementary Table 1: clarified.

Comment 2: Miscalculated the tubular diameter: fully reperform experiments and recalculation.

Comment 3: The irregular and confusing story line: reorganized and rewritten according to reviewer suggestions.

Comment 4: Specificity issues of SCP3 and DDX4 antibodies: antibodies reevaluated and experiments reperformed.

Comment 5: Mechanism interpretation: new descriptions added in the revised manuscript according to the reviewer's comments.

Comment 6: Unclear descriptions led to misunderstanding: reorganized and rewritten to avoid misunderstanding.

Comment 7: Typo of "near": corrected.

Reviewer #2

Comment 1: LCRMP-1 and CRMP-1 immunoblots required: experiments reperformed and new figures added.

Comment 2: Panels of each stage should be the same: corrected and revised.

Comment 3: Higher magnification of LCRMP-1 staining for subcellular localization: experiments reperformed by immunofluorescence staining and confocal scope analysis.

Comment 4: Triple immunostaining with Sertoli cell, F-actin, and DAPI: experiments reperformed by immunofluorescence staining and confocal scope analysis.

Comment 5: LCRMP-1 and CRMP-1 immunoblots required: experiments reperformed and new figures added.

Comment 6: Scenarios for the molecular machinery of compensatory upregulation should be discussed: discussion was reorganized and rewritten to include the possible molecular machinery.

Comment 7: Typo of “eGFP”: corrected.

Reviewer #1 (Remarks to the Author):

This study examined the physiological role of LCRMP-1, which showed strongest expression in the testis, by producing knockout mice and studied the phenotype of them closely. The KO mice showed normal development and growth but the sperm production was significantly reduced. This reduction was then revealed caused by abnormal spermiation. As the LCRMP-1 was supposed to take roles in actin dynamics, authors examined the signaling cascade regarding to F-actin organization to show upregulation of some relating molecules.

This study was performed with sophisticated manner resulting clear and solid data, supporting the conclusion of this manuscript.

I have noticed some minor concerns in the manuscript as described below.

Response:

Thank the reviewer's affirmation. We realize that any concerns from the reviewer aim to improve the quality and reliability of our study. We had revised the manuscript according to the reviewer's comments and suggestions point-by-point in below. Thank you for your professional comments.

1. Line 108, (Supplementary Table 1); In the supplementary Table 1, I'm not sure that each number in the columns represents.

In Supplementary Table 1, it is not clear what each number in the column represents. What pathological scores actually represent is also not clear. The lesions in an organ would be completely different from those in other organs. The score 5 = severe/high is added with 76-100%, which is also confusing.

Response:

We really apologize our unclear and inappropriate illustration of Supplementary Table 1. Actually, regarding the pathology report of mice, it was judged by the veterinarian of Animal Center of College of Medicine, National Taiwan University. The evaluation criteria and interpretation were based on the previous study ¹. However, it was our mistake that we did not clarify each number in the report. The grading levels by pathologists may vary greatly. It should be relatively easy to allocate lesions into severity grades relative to the percentage of the tissue/organ affected. Although the nomenclature used for severity grading varies between laboratories, it appears that the most commonly used grading schemes include four or five severity grades to which either descriptive terms (minimal, mild, moderate, etc) and/or numerical levels (grade 1, 2, 3, etc.) were applied. Therefore, we utilized “0“, “1“, “2“, “3“, “4“, and “5” to represent not present, <1%, 1-25%, 26-50%, 51-75%, and 76-100% of the tissue/organ affected, respectively. In order to disclosure and clarify our evaluation, we fully revised our Supplemental Table 1 with appropriative footnotes and the corresponding description in our manuscript (Line 110-111 and 452-456). Thank you very much.

Revised Supplemental Table 1.

Supplementary Table 1. Histopathological analysis report of major organs in *Lcrmp-1^{+/+}* and *Lcrmp-1^{-/-}* mice.

Major Organ/Tissue	Histopathological findings of nonneoplastic lesions	Grading schemes of pathological finding ^a					
		Lcrmp-1^{-/-} (N=3)			Lcrmp-1^{+/+} (N=3)		
		No. 1	No. 2	No. 3	No. 1	No. 2	No. 3
Brain		0	0	0	0	0	0
Heart		0	0	0	0	0	0

Liver	Extramedullary hematopoiesis, focal ^{b,c}	1	1	1	1	1	1
Kidney	Regeneration, renal tubule, cortex, focal ^d	0	0	0	1	0	0
	Infiltration, mononuclear cell, interstitium, focal	0	1	0	0	0	0
	Mineralization, collecting duct, papilla, focal ^e	0	1	1	1	0	0
Spleen		0	0	0	0	0	0
Lung		0	0	0	0	0	0
Testis	Degeneration, germ cell, seminiferous tubule, focal to diffuse ^f	1	5	5	1	1	1
	Vacuolation, seminiferous tubule, focal ^g	1	1	1	1	1	1
	Atypical residual bodies, seminiferous tubule, focal ^h	1	0	0	1	1	1
	Multinucleated giant cell, seminiferous tubule, focal ⁱ	0	1	1	0	0	0
Sciatic nerve		0	0	0	0	0	0
Pituitary gland		0	0	0	0	0	0
Spinal cord		0	0	0	0	0	0

^aSeverity of lesions was graded according to the previous study ¹. It was relative the percentage of the tissue/organ area affected. The scores, “0” means no presence of significant histology change. Degree of lesions stained with H&E was graded from one to five depending on severity: 1= minimal (<1%); 2: slight (1-25%); 3= moderate (26-50%); 4= moderately severe/high (51-75%); 5= severe/high (76-100%). ^bFocal lesion. ^cThe hematopoietic cells randomly appeared in the hepatic sinusoids. ^dThe necrosis of tubular epithelium followed by regeneration with intact basement membrane. ^eThe lesion with dense basophilic granular deposits as a calcification due to systemic calcium and phosphorus imbalance. ^fThe lesion with decreased elongating spermatids. Germ cell death can be attributed to disturbances in Sertoli cell regulating the survival of germ cell. ^gThe vacuoles within the seminiferous epithelium may result from degenerative changes and the loss of embedded germ cells. ^hThe lesions showed abnormally large or clumped residual bodies, which appeared at the luminal surface or resorbed into Sertoli cell cytoplasm. ⁱThe multinucleated giant cell within the

seminiferous epithelium may result from degeneration of spermatids and spermatocytes.

Line 110-111

However, there were more degenerating germ cells in the testis of *Lcrmp-1^{-/-}* mice than in *Lcrmp-1^{+/+}* mice by histopathological analysis (Supplementary Table 1).

Line 452-456

The H&E staining and histopathological interpretation of mouse organs was performed by the pathologist from the Taiwan Mouse Clinic and Laboratory Animal Center, College of Medicine, National Taiwan University Medical College. Each organ section was examined at lower-power field and quantified with severity grading scheme (Grading scheme I) based on the previous study ¹.

2. Line 120, Fig. 2b; The tubular diameter was measured to be around 500 μm in this experiment. It is strange however that regular mouse seminiferous tubules are having about 200 μm in diameter. I suspect that authors used incorrect measure or miscalculated.

Response:

We deeply apologized this obvious due to incorrect measurements. In previous measurements, we did not pay attention to variations in shape of seminiferous tubules including round and oval caused by cutting plane of sections. To correct this bias, we referred to the previous study to recount ². Briefly, seminiferous tubules only with round or nearly round were chosen randomly and measured for each group. The diameter of the seminiferous tubule was measured across the minor and major axes and the mean diameter was obtained. After recalculation, the diameters of seminiferous tubule cross sections in *Lcrmp-1^{+/+}* and *Lcrmp-1^{-/-}* were $186.51 \pm 23.15 \mu\text{m}$ and $154.93 \pm 18.68 \mu\text{m}$,

respectively. The difference between two genotypes was reached statistical significance ($p < 0.001$). The original Fig. 2b was revised and changed to Fig. 2e according the reviewer's comment in below (revised Fig. 2e) and we had added corresponding description in our revised manuscript (Line 131-134 and 456-461). Thank the reviewer's kindly reminding.

Revised Fig. 2e

Revised Fig. 2

Fig. 2 *Lcrmp-1*^{-/-} mice have fewer spermatozoa and reduced sperm motility.

a The size of testes and epididymides and the ratio of their weights to body weight (n = 7, mean ± SD). **b** The viability of sperm was analyzed using the eosin–nigrosin

staining method. Viability was calculated as the number of white sperm cells divided by the total number of sperm cells (200 sperm cells for each mouse sample; $n = 12$, mean \pm SD). Scale bars, 10 μm . **c** The concentration of spermatozoa in male mice of different ages (6–8 weeks and 6–7 months). The numbers were counted with a hemocytometer ($n = 11$ for 6–8 weeks, $n = 5$ for 6–7 months, mean \pm SD, two-tailed Student's *t*-test, ** $p < 0.01$, *** $p < 0.001$). **d** Sperm motility measurement by OpenCASA software. Three 10 s videos for each mouse were recorded. The trajectories of the spermatozoa were analyzed. White indicates slow trajectories, yellow indicates moderate-speed trajectories, and red indicates fast trajectories. Scale bars, 50 μm . The following parameters of motility were also measured: motility rate, straight-line velocity (VSL), curvilinear velocity (VCL), average-path velocity (VAP), linearity (LIN), straightness (STR), amplitude of lateral head displacement (ALH), beat-cross frequency (BCF), and mean angular displacement (MAD) The original videos of upper panel were showed in Supplementary movie 1. ($n = 9$, mean \pm SD, two-tailed Student's *t*-test, * $p < 0.05$). **e** Left panel: H&E stain of the testes and epididymides of *Lcrmp-1^{-/-}* and *Lcrmp-1^{+/+}* mice. Right panel: diameter of the seminiferous tubules. Seminiferous tubules only with round or nearly round were chosen randomly and measured for each group. The diameters of 30 round tubules from each mouse were measured, and the mean diameters were calculated by averaging the long diameter and short diameter ($n = 6$, mean \pm SD, two-tailed Student's *t*-test, *** $p < 0.001$). Scale bars, 100 μm .

Line 131-134

Hematoxylin–eosin (H&E) staining revealed that the mean diameter of the seminiferous tubules in *Lcrmp-1^{-/-}* mice was also significantly smaller than in *Lcrmp-1^{+/+}* mice ($154.93 \pm 18.68 \mu\text{m}$ v.s $186.51 \pm 23.15 \mu\text{m}$, $p < 0.001$) (Fig. 2e).

Line 456-461

The diameter of seminiferous tubule calculation was according to the previous study². Briefly, only the round or nearly round cross sections of the seminiferous tubules were chosen randomly and measured for each group. The diameter of the seminiferous tubule was measured and averaged across the long and short axes. Totally 180 round tubules from 6 mice each group were obtained for analysis.

3. Line 122 – 135; The story line in this part is irregular and a bit confusing. A simple conclusion seems to be “reduced number of spermatozoa in KO mice” in the sample collected from epididymis. Reduced motility is another abnormal data. Other than those, spermatozoa in epididymis were similar to those in control. Such fact can be written in a simple way.

Response:

We really apologized for the confused story line. Be honest, it was also difficult for us to decide the storytelling in our previous submission. Since the reviewer also had such an opinion, we reorganized and rewrote this paragraph in our manuscript to increase ease of understanding and avoid confusing (revised Fig. 2 and Line 119-136). We can conclude that mice deficient in LCRMP-1 exhibited reduced number of spermatozoa. Please check it. Thank you very much.

Revised Fig. 2

Fig. 2 *Lcrmp-1*^{-/-} mice have fewer spermatozoa and reduced sperm motility.

a The size of testes and epididymides and the ratio of their weights to body weight (n = 7, mean ± SD). **b** The viability of sperm was analyzed using the eosin–nigrosin

staining method. Viability was calculated as the number of white sperm cells divided by the total number of sperm cells (200 sperm cells for each mouse sample; $n = 12$, mean \pm SD). Scale bars, 10 μm . **c** The concentration of spermatozoa in male mice of different ages (6–8 weeks and 6–7 months). The numbers were counted with a hemocytometer ($n = 11$ for 6–8 weeks, $n = 5$ for 6–7 months, mean \pm SD, two-tailed Student's *t*-test, ** $p < 0.01$, *** $p < 0.001$). **d** Sperm motility measurement by OpenCASA software. Three 10 s videos for each mouse were recorded. The trajectories of the spermatozoa were analyzed. White indicates slow trajectories, yellow indicates moderate-speed trajectories, and red indicates fast trajectories. Scale bars, 50 μm . The following parameters of motility were also measured: motility rate, straight-line velocity (VSL), curvilinear velocity (VCL), average-path velocity (VAP), linearity (LIN), straightness (STR), amplitude of lateral head displacement (ALH), beat-cross frequency (BCF), and mean angular displacement (MAD) The original videos of upper panel were showed in Supplementary movie 1. ($n = 9$, mean \pm SD, two-tailed Student's *t*-test, * $p < 0.05$). **e** Left panel: H&E stain of the testes and epididymides of *Lcrmp-1^{-/-}* and *Lcrmp-1^{+/+}* mice. Right panel: diameter of the seminiferous tubules. Seminiferous tubules only with round or nearly round were chosen randomly and measured for each group. The diameters of 30 round tubules from each mouse were measured, and the mean diameters were calculated by averaging the long diameter and short diameter ($n = 6$, mean \pm SD, two-tailed Student's *t*-test, *** $p < 0.001$). Scale bars, 100 μm .

Line 119-136

To elucidate the cause of this reduced fertility, we further evaluated macroscopic and microscopic characteristics including quantity and quality of testes and sperm based on previous histopathological analysis. There were no significant differences in the appearance, weight of the testes, sperm viability, and sperm morphology of *Lcrmp-1^{-/-}*

and *Lcrmp-1^{+/+}* mice (Fig. 2a, b). However, there were substantially fewer spermatozoa in adult *Lcrmp-1^{-/-}*, even in mice aged 6–7 months, this substantial reduction was obvious (Fig. 2c). We further tested the function of the spermatozoa. It was notable that the motility of the sperm in *Lcrmp-1^{-/-}* was significantly lower (Supplementary Movie 1). By using the OpenCASA software analysis, the sperm from *Lcrmp-1^{-/-}* exhibited fewer fast trajectories than *Lcrmp-1^{+/+}* sperm (Fig. 2d). Except for motility, other velocity parameters including curvilinear velocity (VCL), straight-line velocity (VSL), and average-path velocity (VAP) were comparable to those in *Lcrmp-1^{+/+}*. In conclusion, mice deficient in LCRMP-1 exhibited reduced number of spermatozoa. This led us to perform molecular histopathological analysis. Hematoxylin–eosin (H&E) staining revealed that the mean diameter of the seminiferous tubules in *Lcrmp-1^{-/-}* mice was also significantly smaller than in *Lcrmp-1^{+/+}* ($154.93 \pm 18.68 \mu\text{m}$ v.s $186.51 \pm 23.15 \mu\text{m}$, $p < 0.001$) (Fig. 2e). Combined, these results suggested that the low fertility of the *Lcrmp-1^{-/-}* mice was possibly related to the reduced number and activity of spermatozoa, and LCRMP-1 deficiency may have impacts on the process after spermiogenesis.

4. Line 149, Fig. 3c; The staining of SCP3 seems not specific to spermatocytes, but even the interstitial tissue stained also. I think the antibody is not good enough. At closer look, the DDX4 antibody was not so specific to germ cells either.

Response:

Thank the reviewer's reminding. We also noticed this specificity issue. In order to clarify and address whether antibodies conferred to these confusing results, we totally compared additional clones of SCP3 and DDX4. Also, we further adjusted staining procedures especially the antibody dilution buffer. After confirming of antibody issue, we further recalculated positive cell percentage of SCP3 and DDX4 in *Lcrmp-1^{+/+}* and

Lcrmp-1^{-/-} mice of Fig. 3c and 3d (Fig. 3d and 3e in the revised version).

In our original experiments, we used SCP3 and DDX4 antibodies from Abcam (Cat. #ab97672) and Abcam (Cat. #ab13840). With the reviewer's comments, we evaluated additional SCP3 antibody from Genetex (Cat. #GTX15092) and DDX4 antibody from Cell Signaling Technology (Cat. #8761). In conclusion, SCP3 antibody from Cell Signaling Technology (Cat. #8761). In conclusion, SCP3 antibody from Abcam (Cat. #ab97672) and DDX4 antibody from Cell Signaling Technology (Cat. #8761) with Antibody Dilution Buffer (VENTANA, USA) exhibited the most specific patterns and were utilized for revised staining. The staining and quantification results were revised as Fig. 3d and 3e. The proportion of cells that exhibited a positive signal during the early stages of spermatogenesis was significantly higher in *Lcrmp-1*^{-/-} mice than in *Lcrmp-1*^{+/+}. The corresponding description including materials and methods and results (Line 151-155 and 463-477). We highly appreciated the reviewer's carefulness.

Revised Fig. 3d

Revised Fig. 3e

Revised Fig. 3

Fig. 3 LCRMP-1 which is highly expressed near the lumen of seminiferous tubules affects immature germ cells.

a LCRMP-1 expression in the testes, as represented by eGFP, in a Western blotting

analysis of *Lcrmp-1*^{+/+} and *Lcrmp-1*^{-/-} mice. The presence of the eGFP signal confirms that the *Lcrmp-1* gene locus had been replaced by the eGFP gene. **b** Immunohistochemical staining for LCRMP-1 expression patterns in the seminiferous tubules, as indicated by eGFP signaling. LCRMP-1 was highly expressed in the late spermatogenesis. Scale bars, 100 μ m. **c** Immunofluorescence staining for LCRMP-1 localization in the seminiferous tubules using LCRMP-1 specific antibody (green), phalloidin-iFluor (red), and DAPI (blue) followed by confocal microscopy analysis. Scale bars, 20 μ m. **d** SCP3 immunofluorescence staining of the testes of *Lcrmp-1*^{-/-} and *Lcrmp-1*^{+/+} mice. SCP3-positive cells were significantly more abundant in *Lcrmp-1*^{-/-} than in *Lcrmp-1*^{+/+} mice. The fluorescence signal was measured using ImageJ software and the ratio was calculated as the total number of Alexa Fluor 488 Dye (green) cells divided by the total number of DAPI (blue) cells (triplicate, *Lcrmp-1*^{+/+}, n = 5; *Lcrmp-1*^{-/-}, n = 5; mean \pm SD, two-tailed Student's *t*-test, ***p < 0.001). Scale bars, 50 μ m. **e** DDX4 immunofluorescence staining of the testes of *Lcrmp-1*^{-/-} and *Lcrmp-1*^{+/+} mice. DDX4-positive cells were significantly more abundant in *Lcrmp-1*^{-/-} than in *Lcrmp-1*^{+/+} mice. The fluorescence signal was measured using ImageJ software and the ratio was calculated as the total number of Alexa Fluor 488 Dye (green) cells divided by the total number of DAPI (blue) cells (triplicate, *Lcrmp-1*^{+/+}, n = 5; *Lcrmp-1*^{-/-}, n = 5; mean \pm SD, two-tailed Student's *t*-test, **p < 0.01). Scale bars, 50 μ m.

Line 151-155

To test the staging of spermatogenesis in *Lcrmp-1*^{-/-} mice, we identified the spermatogonia and spermatocytes via immunofluorescence staining with the specific antibodies synaptonemal complex protein 3 (SCP3) and DEAD-box helicase 4 (DDX4), respectively (Fig. 3d, e). The proportion of cells that exhibited a positive signal during the early stages of spermatogenesis was significantly higher in *Lcrmp-1*^{-/-} mice than in

Lcrmp-1^{+/+} mice.

Line 463-477

Immunofluorescence staining

For immunofluorescence staining, slides were deparaffinized and retrieved in citrate buffer (pH 6.0) or Tris-EDTA (pH 9.0) at 114–121 °C for 15 min and incubated in Rodent Block M (Biocare Medical, USA) for an hour at room temperature. The slides were incubated in primary antibody (anti-SCP3, Cat. #ab97672, Abcam, 1:100; anti-DDX4, Cat. #8761, Cell Signaling Technology, 1:100, Cat. #ab13840, Abcam, 1:100; anti-LCRMP-1, homemade, 1:100; anti-SOX9, Cat. #H00006662-M01, Abnova, 1:100) diluted with Antibody Dilution Buffer (Ventana, Roche, USA) or 1% BSA phosphate-buffered saline with Tween 20 (PBST) overnight at 4 °C and then in secondary antibody conjugated to Alexa Flour 488 (Cat. #A-11001, Cat. #A-11008, Thermo Fisher Scientific, 1:500) for an hour at room temperature. The samples were stained with Phalloidin (Cat. #ab176757, Abcam, UK) and Hoechst (DAPI) (Cat. #33342, Thermo Fisher Scientific, USA) for an hour and mounted using Fluoromount Aqueous Mounting Medium (Cat. #F4680, Sigma-Aldrich, USA). The results were quantified using ImageJ software. The images were captured using Olympus IX83 and Olympus DP70, and the confocal microscopy images were acquired using Zeiss LSM 880 and Zeiss LSM 780.

5. Line 154;

Here, defects in late spermatogenesis were suspected to be a cause of accumulation of early-stage germ cells. Is it correct to interpret that the decrease in late-stage germ cells created an open space for early-stage germ cells to accumulate? Are there any other possible mechanisms?

Response:

We thank the reviewer for stimulating us to think deeply. The reviewer raised an interesting possibility to interpret our novel finding. We went further to find out whether there is any relevant theory that can support the phenomenon we found. This allows for a comprehensive discussion of the paper.

We agreed the reviewer's opinion that the decrease in late-stage germ cells in *Lcrmp-1*^{-/-} mice (possibly caused by enhanced apoptosis due to defects in late spermatogenesis or spermiation failure [upregulated cleavage caspase-3 expression in *LCRMP-1*^{-/-} mice shown in revised Fig. 7b]) might create an open space for early-stage germ cells to accumulate. According to current data, it is still unclear whether the reduction of tubular diameter in *LCRMP-1*^{-/-} mice conferred to accumulation of early-stage germ cells. Taken together, we had added several sentences in Discussion for enriching the content of our study (Line 283-286). Thank you very much.

Line 283-286

Also, it cannot be excluded that the decrease in late-stage germ cells due to enhanced apoptosis created an open space for early-stage germ cells to accumulate. These resulted in the accumulation of DDX4-positive cells near the adluminal compartment.

6. Line 171 -175; Relating to the above comment, the Fig. 4c showed that number of ES was higher in KO mice than control, while RS and PS were almost same in both groups. Doesn't this result contradict with data in Fig. 3, which insisted increased number of early-stage germ cells, including PS, in KO mice?

Response:

We apologize for the confusing in the description of the manuscript and figures. In Fig. 3, we found DDX4 immature germ cells were significantly increased in

seminiferous tubules of *Lcrmp-1*^{-/-} mice (Fig. 3e). In the mention of heterogenous cell types in seminiferous tubules, to further address the distribution area and impact range of immature germ cells, we utilized BSA density gradient sedimentation for fractionation according to the previous study³. Based on the density, from top to bottom layers, four major layers were classified including elongating spermatids fractions (ES), round spermatids fractions (RS), pachytene spermatocytes fractions (PS), and other fractions (Others). The “ES” fraction means that the fraction should normally and mainly contain more mature spermatids.

In Fig. 4c, we had found that the DDX4-positive immature germ cells were 2.07- and 1.08-fold significantly increased in ES and RS but not PS fractions, respectively, of *Lcrmp-1*^{-/-} mice compared with those of *Lcrmp-1*^{+/+} mice (revised Fig. 4c). Therefore, we suggested that immature germ cells increased in *Lcrmp-1*^{-/-} possibly due to LCRMP-1 deficiency-mediated accumulation of early-stage germ cells. These were consistent with the conclusion of Fig. 3.

To avoid confusing and misunderstanding, we had revised not only the label of Fig. 4c but also the corresponding descriptions (Line 165-180). Thank you for your comments.

Revised Fig. 4c

Revised Fig. 4

Fig. 4 Deficiency of LCRMP-1 mainly expressed in spermatids causes DDX4-positive cells accumulation in the adluminal compartment.

a DDX4 expression pattern in the seminiferous tubules, as revealed by

immunofluorescence staining. Blue, green, and red signals represent the nucleus, DDX4, and phalloidin (F-actin), respectively. P, pachytene spermatocytes; R, round spermatids, E, elongating spermatids. Scale bars, 100 μ m. **b** Expression profiling of LCRMP-1 and DDX4 in fractionated testicular cells from seminiferous tubules. The testicular cells were isolated from the seminiferous tubules in the testes followed by BSA density-gradient fractionation. In total, 28 fractions were collected and 10% of the volume of each fraction was loaded for Western blotting. E1–E4, R1–R5, P1–P2, and O1–O2 represent corresponding elongating spermatids (fractions 1–4), round spermatids (fractions 5–11), pachytene spermatocytes (fractions 12–22), and others (fractions 23–28), respectively. We quantified GFP as an indicator to calculate the relative expression of LCRMP-1, normalized to E1. **c** Immunofluorescence staining of testicular cells from each cell fraction. Blue and green signals represent the nucleus and DDX4, respectively. The fluorescence positive signal ratio was measured using ImageJ. The ratio was calculated as the total number of Alexa Fluor 488 Dye-positive cells divided by the total number of DAPI-positive cells. Each cell fraction was analyzed in biological triplicate (n=3, mean \pm SD, two-tailed Student's *t*-test, ***p* < 0.01; ****p* < 0.001). ES, elongating spermatids; RS, round spermatids; PS, pachytene spermatocytes part. Scale bars, 100 μ m.

Line 165-180

Since there were more DDX4-positive cells in *Lcrmp-1*^{-/-} mice than in the *Lcrmp-1*^{+/+}, we further tested the distribution area and impact range of early-stage germ cells during spermatogenesis. During spermatogenesis, spermatogonia can differentiate into spermatocytes and spermatids as they move from the basal to the adluminal compartment. Based on DDX4 staining and nucleus morphology, we classified specific cell types in different areas, such as pachytene spermatocytes, round spermatids, and

elongating spermatids, according to a previous study ⁴ (Fig. 4a). To precisely analyze and quantify DDX4-positive early-stage germ cells among heterogeneous cell types in seminiferous tubules, we isolated the spermatids and spermatocytes via bovine serum albumin (BSA) density gradient sedimentation for fractionation followed by immunofluorescence staining. We obtained 28 fractions in total, which we further classified into four groups: elongating spermatids fractions (E), round spermatids fractions (R), pachytene spermatocytes fractions (P), and others fractions (O). We found that LCRMP-1 was highly expressed in the spermatids and less so in the spermatocytes (Fig. 4b). The result indicated that DDX4-positive immature germ cells were significantly increased in ES and RS fractions of the *Lcrmp-1*^{-/-} mice compared with those of *Lcrmp-1*^{+/+} mice (Fig. 4c). These results suggest specifically that the lack of LCRMP-1 affected the late stages of spermatogenesis accompanied by the accumulation of early-stage germ cells.

7. Line 216; “nearer” may be near.

Response:

We highly apologize this low-level error. We had corrected it in our revised manuscript (Line 221-224). Thank you for the reminding.

Line 221-224

F-actin was located near the basal compartment, which harbored retained spermatids, in stages VII–VIII, and was less structured near the lumen of seminiferous tubules in stages IX–X, in *Lcrmp-1*^{-/-} mice than in *Lcrmp-1*^{+/+} mice (Fig. 6a).

Reviewer #2 (Remarks to the Author):

Comments for “LCRMP-1 consummates spermiation in male reproduction by stabilizing F-actin organization via PI3K-Akt pathway balancing”.

Chang J-H et al. examined the role of long-form collapsing response mediator protein-1 (LCRMP-1) in the spermatogenesis based on their findings including abundant expression of LCRMP-1 in testis and reduced fertility of LCRMP1-1 knockout (KO) mice. They found that 1) lower motility of LCRMP-1 KO sperm; 2) involvement of LCRMP-1 in the late stages of spermatogenesis; 3) hyperactivation of Akt-mTOR signaling in LCRMP-1 KO testes. They propose LCRMP-1 deficiency causes imbalanced Akt-downstream signaling in turn to bring disorganized F-actin assembly in late spermatogenesis, followed by spermiation failure and germ-cell apoptosis. These findings are interesting and may clarify the molecular mechanism of LCRMP-1-regulated F-actin cytoskeleton in the spermatogenesis.

However, this manuscript is missing several data supporting the role of LCRMP-1 in spermatogenesis including LCRMP-1 protein (not eGFP) expression and subcellular localization and the direct relation of LCRMP-1 and F-actin in spermatids. The molecular machinery of compensatory upregulation of Akt-signaling in the absence of LCRMP-1 is not discussed.

1) Selective expression of LCRMP-1 in testis is shown by mRNA expression (Fig. 1a). This should be confirmed by protein expression of LCRMP-1 and CRMP1 in testes from wild-type and knockout animals by immunoblot analysis.

Response:

We thank the reviewer mentioned this issue. Of course, to confirm the selective expression pattern as shown in RNA level by immunoblot analysis is reasonable and ideal. To be honest, in our original submission, we had tried to confirm the protein expression of LCRMP-1 by Western blot. However, there were objective difficulties including: 1. Low abundant LCRMP-1 expression in all organs (except the brain); 2. The sensitivity of Western blot; 3. The usage of commercial available CRMP-1 antibody which recognized common 497-523 a.a near the C-terminus (Santa Cruz #sc-365345). Therefore, we individually detected mRNA expression of LCRMP-1 and CRMP-1 by specific primers that recognized either exon 1a or exon 1b as an alternative strategy to demonstrate selective expression. In order to meet the reviewer's comment, we tried our hard to reperform Western blot experiments. The result showed that LCRMP-1 was the most abundant in the brain and relative highly expressed in the testis among other organs by LCRMP-1 specific antibody (revised Fig. 1a). The protein expression of LCRMP-1 and CRMP-1 in testis from wild-type and knockout mice was also detected by CRMP-1 antibody. Furthermore, in order to avoid misunderstanding about selective expression, we added the quantification of LCRMP-1 mRNA expression in the brain along with other organs (revised Fig. 1a). The expression of both LCRMP-1 and CRMP-1 in the brain were the highest, up to hundreds of times higher than in other organs. The corresponding description was also added in the manuscript (Line 95-99). We sincerely expected this effort can satisfy the reviewer. We highly appreciated to the reviewer for understanding of the difficulties.

Revised Fig. 1a

Revised Fig. 1

Fig. 1 Expression profiling of LCRMP-1 and generation of LCRMP-1-deficient mice.

a Upper panel: real-time qPCR analysis of LCRMP-1 and CRMP-1 mRNA expression profile and Western blot analysis of LCRMP-1 protein expression profile by LCRMP-1 specific antibody in wild-type (*Lcrmp-1*^{+/+}) mice. Relative expression of qPCR was calculated as target gene/TBP. Lower panel: ratio of the relative mRNA expression

(LCRMP-1/CRMP-1) (n = 6, mean \pm SD, two-tailed Student's *t*-test, * p < 0.05, ** p < 0.01, *** p < 0.001) and Western blot analysis of LCRMP-1 and CRMP-1 protein expression by common CRMP-1 antibody. **b** The targeting strategy for *Lcrmp-1*. Exon 1a of the *Lcrmp-1*^{+/+} allele was replaced with a target vector composed of E1a-eGFP (enhanced green fluorescence protein). Ex1aF and in1aR primer sets were used to detect the *Lcrmp-1*^{+/+} allele, and egfpF and in1aR primer sets for the *Lcrmp-1*^{-/-} allele. **c** Expression levels of LCRMP-1 and CRMP-1 mRNA in brain lysate from *Lcrmp-1*^{+/+}, *Lcrmp-1*^{+/-}, and *Lcrmp-1*^{-/-} mice (n = 3, mean \pm SD). **d** Expression levels of LCRMP-1 and CRMP-1 protein in brain lysate from *Lcrmp-1*^{+/+}, *Lcrmp-1*^{+/-}, and *Lcrmp-1*^{-/-} mice. CRMP-1 has two forms: “Long form” (“L”), known as LCRMP-1, and “short form” (“S”) is the same as CRMP-1. It had a dose-dependent effect on the expression of LCRMP-1. PFC, prefrontal cortex; Hippo, hippocampus; Amy, amygdala; Cere, cerebellum.

Line 95-99

In addition to the brain, we found that among the other major organs, LCRMP-1 is most abundant in the testes in mRNA and protein levels (Fig. 1a, upper panel). In contrast to the other organs, the expression level of LCRMP-1 is higher than that of CRMP-1 (the mRNA ratio of *Lcrmp-1*/*Crmp-1* and protein expression) in the testes, suggesting that it may play an important role in the male reproductive system (Fig. 1a, lower panel).

2) Genotypes of Fig. 5c panels are missing. Multiple panels of the same stages (4 panels of VII-VIII; 2 panels of IX-X; 2 panels of XI-XII) should be explained.

Response:

Thanks for the reviewer's reminding. We apologized this inappropriate illustration that may lead to misunderstanding. We have added the genotype on Fig. 5c (all eGFP

staining in knockout mice). In the mention of panel numbers, actually, we took many photos from each stage. Based on comprehensive observation, LCRMP-1 had the most impact on stage VII-VIII. That was why we showed 2 more panels for VII-VIII. For this comment, we had added more panels for I-III, IV-VI, IX-X and XI-XII stages. The corresponding descriptions were also revised in the revised manuscript (Line 194-205). Thank the reviewer's carefulness.

Revised Fig. 5c

Revised Fig. 5

Fig. 5 LCRMP-1 is associated with spermiation and exhibits a stage-dependent

pattern in the seminiferous tubule stages.

a Statistical analysis of the spermatogenesis stage. Testis sections were classified into 12 stages according to the histological morphology of cross sections of the seminiferous tubules. The 12 stages were further divided into five phases: I–III, IV–VI, VII–VIII, IX–X, and XI–XII. The percentages were calculated by dividing the number of seminiferous tubules in each phase by the total number of tubules analyzed (random fields, *Lcrmp-1*^{-/-}, n = 5, 87 tubules; *Lcrmp-1*^{+/+}, n = 5, 88 tubules). **b** Representative images of each phase. Sections of seminiferous tubules were prepared for H&E staining followed by morphological observation. The arrowhead in stages VII–VIII indicate residual bodies and excess cytoplasm, while the dotted circles in stages IX–XII indicate mature spermatid heads. Scale bars, 20 μm. **c** LCRMP-1 expression pattern during spermatogenesis in cycles of the seminiferous epithelium. Immunohistochemical staining of testis sections was done using eGFP antibodies to reveal the pattern of LCRMP-1 expression. The pattern in each stage was further analyzed. LCRMP-1 expression was cyclical, stage specific, changing according to the stage of the seminiferous tubules. Black arrowheads in the white frame showing a magnified section indicate LCRMP-1 expression colocalized with cytoskeleton-like structures along the dorsal curvature of the spermatid head. Scale bars, 20 μm. **d** The histopathology of stage VII–VIII seminiferous tubules in *Lcrmp-1*^{+/+} and *Lcrmp-1*^{-/-} mice. The black arrowhead in (k) indicates a large cell with a pyknotic nucleus and reddish cytoplasm in the spermatid cell layer. Black arrowheads in (m), (n), (o), and (r) indicate spermatids with excess cytoplasm. Black arrowheads in (q), (r), and (s) indicate spermiation failure or delayed spermiation. The black arrowhead in (t) indicates degenerated spermatids or atypical residual bodies (n = 5 for *Lcrmp-1*^{+/+} and *Lcrmp-1*^{-/-} mice, 22 tubule cross sections showing stages VII–VIII). Scale bar, 20 μm.

Line 194-205

We also analyzed the localization of LCRMP-1 during the cycle of the seminiferous epithelium. Interestingly, there were dynamic changes in the eGFP signal, which represented LCRMP-1 expression in seminiferous tubules of various stages (Fig. 5c). In late stage VIII, spermiation is completed and the preleptotene spermatocytes start transiting the BTB. LCRMP-1 was expressed in spermatocytes (especially primary and secondary) that were distributed in the adluminal area surrounded by the BTB area (Fig. 5c, stage VII-VIII). In stages IX–XII, the expression of LCRMP-1 gradually moved to the spermatids and residual bodies (Fig. 5c, stage IX-X and stage XI-XII). It was strongly expressed in the residual bodies during stages I–VI (Fig. 5c, stage I-III and stage IV-VI). In early stage VII, LCRMP-1 expression gradually “diffused” from the center of the lumen to the basement membrane site (Fig. 5c, stage VII-VIII). At the same time, it was also highly expressed along the dorsal curvature of the spermatid head (Fig. 5c, stage VII-VIII, arrowheads).

3) The expression of LCRMP-1 in seminiferous tubules is shown by eGFP immunohistochemistry (IHC) of LCRMP-1 KO in Fig. 3b. This may be sufficient to show stage-specific LCRMP-1 expression, however, eGFP signal cannot represent the subcellular localization of LCRMP-1. Subcellular localization of LCRMP-1 in developing spermatids along with F-actin-staining should be presented with higher magnification.

Response:

We highly appreciated the reviewer raised this important issue about LCRMP-1 subcellular localization. We agree that eGFP staining of original 3b was insufficient to address this issue. We performed LCRMP-1, phalloidin, and DAPI staining in seminiferous tubules followed by confocal microscope analysis with 63x objective lens

(revised Fig. 3c and original Fig. 3c and 3d were extended to Fig. 3d and 3e, respectively). In addition, we also analyzed this issue in other stages of developing spermatids. The result was responded to other comments of the reviewer (please see below). In revised Fig. 3c, we found that LCRMP-1 was cytoplasmic localization and partially colocalized with phalloidin. The corresponding description was also added in the revised manuscript (Line 148-154). Thanks for the reviewer's reminding.

Note: We also provided an additional immunofluorescence image by confocal microscope with higher magnification (100x objective lens) in this response letter for the reviewer's reference in below. Please check it.

Revised Fig. 3c

Revised Fig. 3

Fig. 3 LCRMP-1 which is highly expressed near the lumen of seminiferous tubules affects immature germ cells.

a LCRMP-1 expression in the testes, as represented by eGFP, in a Western blotting analysis of *Lcrmp-1*^{+/+} and *Lcrmp-1*^{-/-} mice. The presence of the eGFP signal confirms

that the *Lcrmp-1* gene locus had been replaced by the eGFP gene. **b** Immunohistochemical staining for LCRMP-1 expression patterns in the seminiferous tubules, as indicated by eGFP signaling. LCRMP-1 was highly expressed in the late spermatogenesis. Scale bars, 100 μm . **c** Immunofluorescence staining for LCRMP-1 localization in the seminiferous tubules using LCRMP-1 specific antibody (green), phalloidin-iFluor (red), and DAPI (blue) followed by confocal microscopy analysis. Scale bars, 20 μm . **d** SCP3 immunofluorescence staining of the testes of *Lcrmp-1^{-/-}* and *Lcrmp-1^{+/+}* mice. SCP3-positive cells were significantly more abundant in *Lcrmp-1^{-/-}* than in *Lcrmp-1^{+/+}* mice. The fluorescence signal was measured using ImageJ software and the ratio was calculated as the total number of Alexa Fluor 488 Dye (green) cells divided by the total number of DAPI (blue) cells (triplicate, *Lcrmp-1^{+/+}*, n = 5; *Lcrmp-1^{-/-}*, n = 5; mean \pm SD, two-tailed Student's *t*-test, ***p < 0.001). Scale bars, 50 μm . **e** DDX4 immunofluorescence staining of the testes of *Lcrmp-1^{-/-}* and *Lcrmp-1^{+/+}* mice. DDX4-positive cells were significantly more abundant in *Lcrmp-1^{-/-}* than in *Lcrmp-1^{+/+}* mice. The fluorescence signal was measured using ImageJ software and the ratio was calculated as the total number of Alexa Fluor 488 Dye (green) cells divided by the total number of DAPI (blue) cells (triplicate, *Lcrmp-1^{+/+}*, n = 5; *Lcrmp-1^{-/-}*, n = 5; mean \pm SD, two-tailed Student's *t*-test, **p < 0.01). Scale bars, 50 μm .

Line 148-154

We further characterized subcellular localization of LCRMP-1 in the seminiferous tubules by co-staining with LCRMP-1, phalloidin, and DAPI (Fig. 3c). The result showed that LCRMP-1 was majorly cytoplasmic localization and partially colocalized with phalloidin. To test the staging of spermatogenesis in *Lcrmp-1^{-/-}* mice, we identified the spermatogonia and spermatocytes via immunofluorescence staining with the specific antibodies synaptonemal complex protein 3 (SCP3) and DEAD-box

helicase 4 (DDX4), respectively (Fig. 3d, e).

100x magnified immunofluorescence image

4) It is difficult to image the illustrated F-actin distribution (Fig. 6a right panels) from the Phalloidin staining (Fig. 6a left and middle panels). Higher magnification view is required (inset pictures are insufficient). Triple immunostaining with Sertoli cell marker may be examined to show the different distribution of F-actin in LCRMP-1 KO seminiferous tubules.

Response:

Thank the reviewer's reminding. We totally agreed that it was difficult to image the illustrated F-actin distribution only based on the staining of original Fig. 6a left and middle panels. The low power field was not sufficient to fully support our description. To further provide solid evidences, we had reformed experiments of triple immunostaining with Sertoli cell marker, SOX9, Phalloidin, and DAPI followed by confocal microscopy analysis under 100x magnification view according to the reviewer's recommendation (revised Fig. 6b).

During spermatogenesis, the actin network supports cytoskeleton structure, the apical ectoplasmic specializations accelerating the separation of mature spermatid heads from the Sertoli cells in stages VII-VIII. The actin filaments and actin dynamics are required for the elongating spermatids in acrosome formation and flagella elongation phases near the lumen of seminiferous tubules in stages IX-X.

Our results showed that F-actin was well-organized forming a nest-like or fence-like structure near the dorsal of spermatid head during spermiation in *Lcrmp-1*^{+/+} mice in stages VII-VIII. These organized structures promote the removal of the residual bodies from spermatids and enhance the successful spermiation. Compared with the *Lcrmp-1*^{+/+} mice, the *Lcrmp-1*^{-/-} showed that F-actin appeared wrapping around heads and parts of tails of spermatids in stages VII-VIII. In addition, in stages IX-X, the condensed and bundled F-actin appeared surrounding the tails of elongating spermatids in *Lcrmp-1*^{+/+} mice while the diffuse and disrupt F-actin organizations were observed in *Lcrmp-1*^{-/-} (revised Fig. 6b). The results agree with our previous data (Fig 5c), which showed LCRMP-1 localizing near the spermatid head regulating actin dynamics in the late spermatogenesis. In addition, with Sertoli cell marker (SOX9), we can observe the localization of the Sertoli cells, F-actin network, and spermatid heads at the same time. The cytoplasm of the Sertoli cell extends from basal compartment to adluminal compartment, and F-actin with cytoskeleton structures are mainly distributed between nuclei of Sertoli cell and spermatid head accelerating the differentiation and separation of spermatids.

These results further support the illustration of original Fig. 6a right panel. We highly appreciated the reviewer guided us to improve our results. To describe our results more clearly, we reorganized the order of figures. We combined the new triple immunostaining (revised Fig. 6b) with original Fig. 6a left panels (revised Fig. 6a) and original Fig. 6a right panels (revised Fig. 6c) as Fig. 6. The original Fig. 6b and 6c

(Western blot analysis) were shifted to the new Fig. 7 (new Fig. 7). This rearrangement aims to separate storytelling for different types of results. The corresponding descriptions were added and changed in our revised manuscript (Line 225-235). Thank the reviewer's understanding.

Revised Fig. 6b

Revised Fig. 6

Fig. 6 Disorganized F-actin in *Lcrmp-1^{-/-}* mice.

a Testis sections were stained using DAPI and phalloidin and subjected to confocal microscope analysis. Green arrowheads indicate retained spermatids near the basal compartment, while the white arrows in the white frames showing magnified sections indicate mature elongated spermatids at stages VII–VIII. In stages XI–X, the white frames show magnified images of the elongating spermatids near the lumen of the seminiferous tubules. The phalloidin results were shown as grayscale images. Scale

bars, 20 μm . **b** Immunofluorescence staining of SOX9 (Sertoli cell), phalloidin (F-actin), and DAPI in stage VII-VIII and IX-X of *Lcrmp-1^{+/+}* and *Lcrmp-1^{-/-}* mice seminiferous tubules followed by confocal microscope analysis. Arrowheads in Stage VII-VIII indicate F-actin wrapping around mature spermatids, and arrowheads in IX-X show disorganized F-actin in elongating spermatids. Scale bars, 20 μm . **c** Illustration of disorganized F-actin in *Lcrmp-1^{-/-}* mice in comparison to F-actin in wild-type mice. The filamentous F-actin retains mature spermatids near the basal compartment of the seminiferous epithelium and surrounds the tails and heads of spermatids during separation from the Sertoli cells in stages VII–VIII as shown in 6a, b. The disorganized F-actin appears near the tails of elongating spermatids in stages IX–X in *Lcrmp-1^{-/-}* mice as shown in 6a, b.

Revised Fig. 7

Fig. 7 Highly phosphorylated upstream molecules of LCRMP-1 in *Lcrmp-1*^{-/-} mice.

a Western blotting analysis and quantification of whole-testis lysate for PI3K–Akt–mTOR/GSK3β signaling related to F-actin dynamics in *Lcrmp-1*^{+/+} and *Lcrmp-1*^{-/-} mice (n = 5, mean ± SD, two-tailed Student's *t*-test, * p < 0.05, ** p < 0.01, *** p < 0.001). **b** Western blotting analysis and quantification of fractionated testicular cells (spermatids, spermatocytes, and others) of *Lcrmp-1*^{+/+} and *Lcrmp-1*^{-/-} mice. HEF

indicates pooled cell fractions 1–8, which exhibit high expression of LCRMP-1, while LEF indicates pooled cell fractions 9–20, which exhibit low expression of LCRMP-1 (n = 3, mean ± SD, two-tailed Student's *t*-test, * p < 0.05, ** p < 0.01, *** p < 0.001).

Line 225-235

To further characterize the organization of F-actin in *Lcrmp-1*^{-/-} mice, immunofluorescence triple staining of SOX9 (Sertoli cell), phalloidin (F-actin), and DAPI followed by confocal microscope analysis was performed (Fig. 6b). The results showed that F-actin was well-organized forming a nest-like or fence-like structure between the dorsal of spermatid head and the nuclei of the Sertoli cell that facilitated the removal of the residual bodies from spermatids in *Lcrmp-1*^{+/+} mice in stages VII–VIII while F-actin appeared wrapping around the heads and parts of tails of spermatid in *Lcrmp-1*^{-/-} (Fig. 6b, arrowheads). In addition, in stages IX–X, the condensed and bundled F-actin appeared surrounding the tails of elongating spermatids in *Lcrmp-1*^{+/+} mice while the diffuse and disrupt F-actin organizations were observed in *Lcrmp-1*^{-/-} (Fig. 6b, arrowheads). These results indicated that LCRMP-1 deficiency caused disorganization of F-actin resulting in aberrant spermiation (Fig. 6c).

5) Fig. 6b and 6c panels should include LCRMP-1 and CRMP1 immunoblots.

Response:

Thank the reviewer's reminding. We apologized this carelessness. We sincerely hoped that we can obtain the understanding of the reviewer that this experiment had objective difficulties due to either the trace amount of LCRMP-1 in testis fractions or high background noise and interference in BSA fractionation. Even so, we still tried our best to perform this experiment to meet the reviewer's requirement. We hoped this can satisfy the reviewer. With the consideration of other comments, we had rearranged the

order of figures. Original Fig. 6b and 6c had been changed to Fig. 7a and 7b (revised Fig. 7) and corresponding descriptions were also added in the revised manuscript (Line 237-261).

Revised Fig. 7

Fig. 7 Highly phosphorylated upstream molecules of LCRMP-1 in *Lcrmp-1*^{-/-} mice.

A Western blotting analysis and quantification of whole-testis lysate for PI3K–Akt–

mTOR/GSK3 β signaling related to F-actin dynamics in *Lcrmp-1^{+/+}* and *Lcrmp-1^{-/-}* mice (n = 5, mean \pm SD, two-tailed Student's *t*-test, * p < 0.05, ** p < 0.01, *** p < 0.001). **b** Western blotting analysis and quantification of fractionated testicular cells (spermatids, spermatocytes, and others) of *Lcrmp-1^{+/+}* and *Lcrmp-1^{-/-}* mice. HEF indicates pooled cell fractions 1–8, which exhibit high expression of LCRMP-1, while LEF indicates pooled cell fractions 9–20, which exhibit low expression of LCRMP-1 (n = 3, mean \pm SD, two-tailed Student's *t*-test, * p < 0.05, ** p < 0.01, *** p < 0.001).

Line 237-261

The imbalanced Akt-mTOR-p70S6K/GSK3 β axis interferes F-actin dynamics during spermatogenesis in mice deficient in LCRMP-1

To further understand the mechanism by which LCRMP-1 mediates the process of spermiation, we focused on cytoskeletal arrangements in the late stages of spermatogenesis. Based on previous studies, we hypothesized that the F-actin dynamics and organization required for normal spermiation can be stabilized by either ribosomal protein S6 kinase (p70S6K) or LCRMP-1, regulated by protein kinase B (Akt)-downstream mammalian target of rapamycin (mTOR) and glycogen synthase kinase-3 beta (GSK3 β), respectively ^{5,6}. Based on a previous finding that Akt is a key molecule regulating the ectoplasmic specialization dynamics in the seminiferous tubules at various stages ⁷, we first tested the signaling involved in Akt-mediated F-actin dynamics (Fig. 7a). The expression of LCRMP-1 as control, compared with *Lcrmp-1^{+/+}* mice, phosphorylated Akt (p-Akt) and upstream phosphorylated phosphoinositide 3-kinase (p-PI3K) signaling were significantly upregulated in the testes of *Lcrmp-1^{-/-}* mice. The expression of downstream molecules, p-mTOR and p-GSK3 β , was also enhanced as a result of Akt activation. Furthermore, we observed increased levels of cleaved caspase 3 in the testes of *Lcrmp-1^{-/-}* mice, suggesting that germ-cell apoptosis

may be the consequence of spermiation failure. Considering all the effects on heterogenous cell populations in the testis that are caused by LCRMP-1 deficiency, we separated testicular cells into high- and low-LCRMP-1-expressing fractions (HEF and LEF, respectively), based on LCRMP-1 and eGFP expression, for further validation (Fig. 7b). We found that p-Akt, p-mTOR, p-GSK3 β , and p-p70S6K were upregulated in the LEF of *Lcrmp-1*^{-/-} mice, while cleaved caspase 3 was significantly upregulated in their HEF. Combined, these results suggest that LCRMP-1 deficiency causes F-actin disorganization during late spermatogenesis, followed by spermiation failure and germ-cell apoptosis. LCRMP-1 deficiency may cause imbalanced regulation by two Akt-downstream signaling pathways, enhancing the mTOR-p70S6K signal and resulting in the F-actin disorganization phenotype in stages VII-X (Supplementary Fig. 6).

6) Several scenarios for the molecular machinery of compensatory upregulation of Akt-signaling in the absence of LCRMP-1 may be proposed in the discussion.

Response:

Thank the reviewer's reminding. We will add several scenarios to mention the molecular machinery which may be involved in compensatory upregulation of Akt-signaling in the absence of LCRMP-1.

In previous studies, CRMPs proteins were known as cargo adaptors connecting kinesin with cargo proteins such as Sra-1/WAVE-1 complexes or interacting with factors promoting polymerization regulating axon formation or cell adhesion.⁸⁻¹¹ It is possible that LCRMP-1 may serve as either a cargo adaptor transporting the Sra-1/WAVE1 or Arp2/3 complexes toward the apical ES and TBC for supporting assembly and recycle of actin-based junctional structures or a molecule binding with actin-elongation factors to facilitate actin polymerization and remodeling in adluminal compartments during spermiation. LCRMP-1 may promote the separation of sperm

from Sertoli cell through accelerating the conversion from the bundled actin microfilaments to branched actin filaments¹². Therefore, under the condition of LCRMP-1 deficiency, the transportation of these complexes regulating actin assembly and stability becomes less efficiency leading to reduced rates of actin polymerization and imbalanced F-actin/G-actin dynamics. In order to rescue the unstable F-actin organization and maintain the homeostasis of actin dynamics, the upregulated phosphorylation of PI3K-Akt-mTOR-p70S6K pathway compensates this situation by inhibition of F-actin depolymerization through regulating cofilin^{5,13,14}. Furthermore, Ser9 phosphorylation of GSK3 β can be an active form to phosphorylate Gli3 downregulating the Hedgehog pathway, which may be regulated by actin cytoskeleton and modulate cytoskeleton remodeling in axon guidance and contain a negative feedback loop base on previous studies¹⁵⁻²⁰. From our perspective, without interaction between GSK3 β and LCRMP-1 in *Lcrmp-1*^{-/-}, the level of phosphorylated GSK3 β (Ser9) is increased and may induce feedback loop to regulate the Hedgehog pathway promoting actin remodeling. We further presumed that the level of phosphorylation of GSK3 β may affect PI3K-Akt-mTOR-p70S6K pathway through the Hedgehog pathway on the basis of previous studies^{15,21}. However, this complicated network needs to be further investigated. Consequently, the loss of LCRMP-1 causes reduced actin polymerization and stability leading to disorganization in spite of compensation by inhibiting actin depolymerization through upregulated Akt-signaling.

We had added these possible scenarios in the discussion to rich the content of manuscript (Line 317-337). According to these inputs, we had revised and reorganize whole Discussion to avoid redundance. Please check it. Thank you very much.

Line317-337

Previous studies showed CRMPs served as either a cargo adaptor transporting the Sra-

1/WAVE1 and Arp2/3 complexes toward the adluminal compartments for supporting assembly and recycle of actin-based junctional structures or a molecule binding with actin-elongation factors to facilitate actin polymerization and remodeling during spermiation⁸⁻¹¹. LCRMP-1 may promote the separation of sperm from Sertoli cell through accelerating the conversion from the bundled actin microfilaments to branched actin filaments¹². Therefore, under the condition of LCRMP-1 deficiency, the transportation of these complexes regulating actin assembly and stability becomes less efficiency leading to reduced rates of actin polymerization and imbalanced F-actin/G-actin dynamics. In order to rescue the unstable F-actin organization and maintain the homeostasis of actin dynamics, the upregulated phosphorylation of PI3K-Akt-mTOR-p70S6K pathway compensates this situation by inhibition of F-actin depolymerization through regulating cofilin^{5,13,14}. Furthermore, based on previous studies, Ser9 phosphorylation of GSK3 β can be an active form to phosphorylate Gli3 downregulating the Hedgehog pathway^{15,16}. The Hedgehog pathway may be regulated by actin cytoskeleton and PI3K-Akt-mTOR-p70S6K pathway, and modulate cytoskeleton reorganization in axon guidance, and contain a negative feedback loop¹⁷⁻²¹. From our perspective, without the interaction between GSK3 β and LCRMP-1 in *Lcrmp-1*^{-/-} mice, the level of phosphorylated GSK3 β (Ser9) is increased, which may induce feedback loop to regulate the Hedgehog pathway promoting actin remodeling and affect the PI3K-Akt-mTOR-p70S6K pathway. However, this complicated network needs to be further investigated. In conclusion, LCRMP-1 deficiency exhibits a homeostatic imbalance of the two pathways leads to unstable F-actin dynamics and spermiation abnormalities.

7) L237, eGFR should be eGFP

Response:

We highly apologize this error. We had corrected it in our revised manuscript (Line 252-255). Thank you for the reminding.

Line 252-255

Considering all the effects on heterogenous cell populations in the testis that are caused by LCRMP-1 deficiency, we separated testicular cells into high- and low-LCRMP-1–expressing fractions (HEF and LEF, respectively), based on eGFP expression, for further validation (Fig. 7b).

References:

- 1 Shackelford, C., Long, G., Wolf, J., Okerberg, C. & Herbert, R. Qualitative and quantitative analysis of nonneoplastic lesions in toxicology studies. *Toxicol Pathol* **30**, 93-96, doi:10.1080/01926230252824761 (2002).
- 2 Tripathi, U. K. *et al.* Morphometric evaluation of seminiferous tubule and proportionate numerical analysis of Sertoli and spermatogenic cells indicate differences between crossbred and purebred bulls. *Vet World* **8**, 645-650, doi:10.14202/vetworld.2015.645-650 (2015).
- 3 Da Ros, M., Lehtiniemi, T., Olotu, O., Meikar, O. & Kotaja, N. Enrichment of Pachytene Spermatocytes and Spermatids from Mouse Testes Using Standard Laboratory Equipment. *J Vis Exp*, doi:10.3791/60271 (2019).
- 4 Isoler-Alcaraz, J., Fernandez-Perez, D., Larriba, E. & Del Mazo, J. Cellular and molecular characterization of gametogenic progression in ex vivo cultured prepuberal mouse testes. *Reprod Biol Endocrinol* **15**, 85, doi:10.1186/s12958-017-0305-y (2017).
- 5 Ip, C. K. & Wong, A. S. p70 S6 kinase and actin dynamics: A perspective. *Spermatogenesis* **2**, 44-52, doi:10.4161/spmg.19413 (2012).
- 6 Crouch, M. F. Regulation of thrombin-induced stress fibre formation in Swiss 3T3 cells by the 70-kDa S6 kinase. *Biochem Biophys Res Commun* **233**, 193-199, doi:10.1006/bbrc.1997.6419 (1997).
- 7 Siu, M. K., Wong, C. H., Lee, W. M. & Cheng, C. Y. Sertoli-germ cell anchoring junction dynamics in the testis are regulated by an interplay of lipid and protein kinases. *J Biol Chem* **280**, 25029-25047, doi:10.1074/jbc.M501049200 (2005).
- 8 Kawano, Y. *et al.* CRMP-2 is involved in kinesin-1-dependent transport of the Sra-1/WAVE1 complex and axon formation. *Mol Cell Biol* **25**, 9920-9935, doi:10.1128/MCB.25.22.9920-9935.2005 (2005).

- 9 Pan, S. H. *et al.* The ability of LCRMP-1 to promote cancer invasion by enhancing filopodia formation is antagonized by CRMP-1. *J Clin Invest* **121**, 3189-3205, doi:10.1172/JCI42975 (2011).
- 10 Yu-Kemp, H. C., Kemp, J. P., Jr. & Briehar, W. M. CRMP-1 enhances EVL-mediated actin elongation to build lamellipodia and the actin cortex. *J Cell Biol* **216**, 2463-2479, doi:10.1083/jcb.201606084 (2017).
- 11 Namba, T., Nakamuta, S., Funahashi, Y. & Kaibuchi, K. The role of selective transport in neuronal polarization. *Dev Neurobiol* **71**, 445-457, doi:10.1002/dneu.20876 (2011).
- 12 Qian, X. *et al.* Actin binding proteins, spermatid transport and spermiation. *Semin Cell Dev Biol* **30**, 75-85, doi:10.1016/j.semcd.2014.04.018 (2014).
- 13 Ip, C. K., Cheung, A. N., Ngan, H. Y. & Wong, A. S. p70 S6 kinase in the control of actin cytoskeleton dynamics and directed migration of ovarian cancer cells. *Oncogene* **30**, 2420-2432, doi:10.1038/onc.2010.615 (2011).
- 14 Li, J. *et al.* NCAM regulates the proliferation, apoptosis, autophagy, EMT, and migration of human melanoma cells via the Src/Akt/mTOR/cofilin signaling pathway. *J Cell Biochem* **121**, 1192-1204, doi:10.1002/jcb.29353 (2020).
- 15 Trnski, D. *et al.* GSK3beta and Gli3 play a role in activation of Hedgehog-Gli pathway in human colon cancer - Targeting GSK3beta downregulates the signaling pathway and reduces cell proliferation. *Biochim Biophys Acta* **1852**, 2574-2584, doi:10.1016/j.bbadis.2015.09.005 (2015).
- 16 Glibo, M. *et al.* The role of glycogen synthase kinase 3 (GSK3) in cancer with emphasis on ovarian cancer development and progression: A comprehensive review. *Bosn J Basic Med Sci* **21**, 5-18, doi:10.17305/bjbms.2020.5036 (2021).
- 17 Schneider, P. *et al.* Identification of a novel actin-dependent signal transducing module allows for the targeted degradation of GLI1. *Nat Commun* **6**, 8023,

- doi:10.1038/ncomms9023 (2015).
- 18 Yam, P. T., Langlois, S. D., Morin, S. & Charron, F. Sonic hedgehog guides axons through a noncanonical, Src-family-kinase-dependent signaling pathway. *Neuron* **62**, 349-362, doi:10.1016/j.neuron.2009.03.022 (2009).
- 19 Carballo, G. B., Honorato, J. R., de Lopes, G. P. F. & Spohr, T. A highlight on Sonic hedgehog pathway. *Cell Commun Signal* **16**, 11, doi:10.1186/s12964-018-0220-7 (2018).
- 20 Lai, K., Robertson, M. J. & Schaffer, D. V. The sonic hedgehog signaling system as a bistable genetic switch. *Biophys J* **86**, 2748-2757, doi:10.1016/S0006-3495(04)74328-3 (2004).
- 21 Flemban, A. & Qualtrough, D. The Potential Role of Hedgehog Signaling in the Luminal/Basal Phenotype of Breast Epithelia and in Breast Cancer Invasion and Metastasis. *Cancers (Basel)* **7**, 1863-1884, doi:10.3390/cancers7030866 (2015).

Reviewers' comments:

Reviewer #1 (Remarks to the Author):

I appreciate that authors responded positively to my comments and the manuscript was improved from my point of view.

There are yet several points that I would like to comment on, as shown below.

Line 31-33: "In conclusion, LCRMP-1 maintains homeostasis of spermatogenesis by consummating spermiation, and the *Lcrmp-1-/-* mouse model provides a potential strategy for investigating male reproductive system."

This last sentence in the abstract seems a bit out of focus and not appropriate as summarizing the study. The latter half of the sentence, in particular, is questionable how the *Lcrmp-1-/-* mouse model can provide a strategy for investigating such a huge research area of male reproductive system.

Line 177: "The result indicated that DDX4-positive immature germ cells were significantly increased in ES and RS fractions of the *Lcrmp-1-/-* mice compared with those of *Lcrmp-1+/+* mice (Fig. 4c)." The increase of DDX4-positive immature germ cells in ES and RS fractions seems to me indicating that the DDX4-positive immature germ cells in *Lcrmp-1-/-* mice were somehow trapped more preferentially in ES and RS fractions than in *Lcrmp-1+/+* mice. If this interpretation is correct, how could this have happened? Did they have a greater or lesser specific gravity, which is same as ES or RS? Otherwise, is it possible that the ES and RS in *Lcrmp-1-/-* mice might have maintained the expression of DDX4? In fact, in the discussion, in line 278-280, authors wrote as follows. "In particular, the spermatids in which LCRMP-1 should have been highly expressed exhibited significantly higher DDX4 positivity." This sentence seems to mean that the spermatids themselves changed to express DDX4 more strongly in *Lcrmp-1-/-* mice than *Lcrmp-1+/+* mice.

To begin with, what are these DDX4-positive immature germ cells exactly? Are they spermatogonia? I strongly feel that there are confusion in my understanding or in the manuscript. Some readers may feel in the same way.

Line 170: The reference 23 is about the in vitro spermatogenesis using an organ culture method. In this reference paper I did not find DDX4 or VASA. I suspect that the reference is incorrect.

Line 185: "In mice, the complete process of spermatogenesis takes about 8.6 days, and different stages of spermatogenesis occur in each segment of the seminiferous tubules."

The complete process of spermatogenesis in mice takes 35 days, not 8.6 days. The 8.6 days are a period taken by a single cycle of the seminiferous epithelium takes place.

Line 283-286; Although authors have taken my comments in the revised manuscript of this part, "an open space" hypothesis may be inappropriate here. I'm sorry but now I think this sentence may be better to be omitted, if authors agree.

Reviewer #2 (Remarks to the Author):

Comments for revised manuscript of "LCRMP-1 consummates spermiation in male reproduction by stabilizing F-actin organization via PI3K-Akt pathway balancing".

Very little is known about the molecular mechanism of spermatogenesis. Chang J-H et al. revealed

that long-form collapsing response mediator protein-1 (LCRMP-1) is involved in the spermatogenesis based on their findings including abundant expression of LCRMP-1 in testis and reduced fertility of LCRMP1 knockout (KO) mice. They found that LCRMP-1 is localized in spermiation machinery at the late stages of spermatogenesis. Using the LCRMP-1 KO mice, they showed that the absence of LCRMP-1 brings disorganized F-actin assembly in the late spermatogenesis, spermiation failure and germ-cell apoptosis, and the upregulation of Akt-mTOR signaling. In this revised manuscript, they discussed possible molecular mechanisms connecting LCRMP-1 and spermiation through Akt-mTOR pathway. The authors have earnestly addressed all of the comments raised by reviewers #1 and #2. I am satisfied with their revisions and find these results very interesting. This study becomes as a significant contribution.

Minor point

Fig.5c, Lcrmp1 -/- (left side label) is missing.

Response summary

The following letter contains full point-by-point responses to both reviewers (5 from the reviewer #1 and 1 from the reviewer #2). We briefly summarized responses in below with a hyperlink to each detail description within the letter.

For the convenience of reviewers to read, we additionally provided a clear manuscript file with line number labeled as the supplemental information. The line number in the response letter was based on this supplementary file.

Reviewer #1

Comment 1: Out of focus in summary: rewritten

Comment 2: Confusion in interpretation of the DDX4-expression data in fractions: reorganized and rewritten

Comment 3: Insufficient reference: additional reference added

Comment 4: Incorrect descriptions: corrected and rewritten

Comment 5: Inappropriate descriptions: omitted

Reviewer #2

Comment 1: Label missing: revised

Reviewer #1 (Remarks to the Author):

I appreciate that authors responded positively to my comments and the manuscript was improved from my point of view. There are yet several points that I would like to comment on, as shown below.

1. Line 31-33: “In conclusion, LCRMP-1 maintains homeostasis of spermatogenesis by consummating spermiation, and the *Lcrmp-1*^{-/-} mouse model provides a potential strategy for investigating male reproductive system.” This last sentence in the abstract seems a bit out of focus and not appropriate as summarizing the study. The latter half of the sentence, in particular, is questionable how the *Lcrmp-1*^{-/-} mouse model can provide a strategy for investigating such a huge research area of male reproductive system.

Response:

Thank the reviewer’s comment. We agree with the reviewer that “the *Lcrmp-1*^{-/-} mouse model provides a potential strategy for investigating male reproductive system” may be suspected of a little bit overinterpretation. Also, this statement may be out of focus and not appropriate as summarizing our finding. Therefore, we had modified the last sentence as “In conclusion, LCRMP-1 maintains homeostasis of spermatogenesis by consummating spermiation by modulating cytoskeleton remodeling for spermatozoa release” (Line 30-32). Thank you.

Line 30-32

In conclusion, LCRMP-1 maintains homeostasis of spermatogenesis by consummating spermiation by modulating cytoskeleton remodeling for spermatozoa release.

2. Line 177: “The result indicated that DDX4-positive immature germ cells were significantly increased in ES and RS fractions of the *Lcrmp-1^{-/-}* mice compared with those of *Lcrmp-1^{+/+}* mice (Fig. 4c).” The increase of DDX4-positive immature germ cells in ES and RS fractions seems to me indicating that the DDX4-positive immature germ cells in *Lcrmp-1^{-/-}* mice were somehow trapped more preferentially in ES and RS fractions than in *Lcrmp-1^{+/+}* mice. If this interpretation is correct, how could this have happened? Did they have a greater or lesser specific gravity, which is same as ES or RS? Otherwise, is it possible that the ES and RS in *Lcrmp-1^{-/-}* mice might have maintained the expression of DDX4? In fact, in the discussion, in line 278-280, authors wrote as follows. “In particular, the spermatids in which LCRMP-1 should have been highly expressed exhibited significantly higher DDX4 positivity.” This sentence seems to mean that the spermatids themselves changed to express DDX4 more strongly in *Lcrmp-1^{-/-}* mice than *Lcrmp-1^{+/+}* mice. To begin with, what are these DDX4-positive immature germ cells exactly? Are they spermatogonia? I strongly feel that there are confusion in my understanding or in the manuscript. Some readers may feel in the same way.

Response:

Thank the reviewer’s comments. We agreed with the reviewer that many readers may be confused to our description and raise the same challenge. And we apologize for the confusion. It should be noticed that BSA density isolation allowed cells to be separated based on specific gravity. Therefore, in normal circumstances, the mature and immature germ cells should be localized to corresponding fractions (layers). How could the DDX4-positive immature germ cells in *Lcrmp-1^{-/-}* mice have been “trapped” in ES and RS fractions especially the ES fraction? We think there are three possibilities. One is the number of DDX4-positive immature germ cells, which were spermatogonia, spermatocytes, and early spermatids¹, increased in *Lcrmp-1^{-/-}* mice so that some cells overflowed into ES and RS fractions. Another is that the differentiation was interfered by increased immature germ cells due to LCRMP-1 deficiency so that each stage cells stacking was out of original order with alterations of gravity (density). The other is that, as mentioned by the reviewer, the expression of DDX4 might be maintained in late-stage spermatids due to LCRMP-1 deficiency. However, without further evidences, we prefer the first possibility based on: 1. The morphology of DAPI staining indicated DDX4-positive cells were similar to immature germ cells; 2. On the basis of LCRMP-1 characteristics, it may not affect the DDX4 expression directly.

Taken together, with the reviewer clarification, we agreed that the present description may lead to some confusions and misunderstandings. We had corrected our sentences in the revised manuscript (Line 279-282). Thank you very much.

Line 279-282

In particular, the presence of more DDX4-positive immature germ cells in the fractions where LCRMP-1 should have been highly expressed suggested that the accumulation of these cells overflowed into these fractions.

3. Line 170: The reference 23 is about the in vitro spermatogenesis using an organ culture method. In this reference paper I did not find DDX4 or VASA. I suspect that the reference is incorrect.

Response:

We apologized our unclear description. The reference 23 was cited due to the guidance for us to differentiate the different stages of germ cells by different nuclear morphology. Our apologies, we did not cite the reference related to DDX4-expression pattern for identification of different germ cells at the end of this sentence. We added additional references and please check it out (Line 167-170). Thank you so much!

Line 167-170

Based on DDX4 staining and nucleus morphology, we differentiated stages and classified specific cell types in different areas, such as spermatogonia, pachytene spermatocytes, round spermatids, and elongating spermatids, according to a previous study¹⁻³ (Fig. 4a).

4. Line 185: “In mice, the complete process of spermatogenesis takes about 8.6 days, and different stages of spermatogenesis occur in each segment of the seminiferous tubules.” The complete process of spermatogenesis in mice takes 35 days, not 8.6 days. The 8.6 days are a period taken by a single cycle of the seminiferous epithelium takes place.

Response:

We thank the reviewer’s correction. We are sorry for our carelessness and mistake. We have corrected the sentence (Line 185-189). Thanks for the reviewer’s kindly reminding.

Line 185-189

In mice, the complete process of spermatogenesis takes about 35 days, which is four times longer than the period of a single seminiferous epithelial cycle from stage I to stage XII (8.6 days), and different stages of spermatogenesis occur in each segment of the seminiferous tubules. This pattern is called the “cycle of the seminiferous epithelium”, which is divided into 12 stages (I–XII)⁴⁻⁶.

5. Line 283-286; Although authors have taken my comments in the revised manuscript of this part, “an open space” hypothesis may be inappropriate here. I’m sorry but now I think this sentence may be better to be omitted, if authors agree.

Response:

Thank the reviewer’s understanding. Actually, we appreciated that the reviewer raised this hypothesis previously. It was interesting for us and promoted us to have in-depth thinking. Since we had no further evidence or data to support this hypothesis, we agreed with the reviewer that we omitted this discussion at this moment (Line 284-285). We highly appreciated the reviewer’s inputs.

Line 284-285

It is possible that numbers of meiotic and early-stage germ cells, which are DDX4 positive, accumulated due to abnormal spermiation in *Lcrmp-1^{-/-}* mice. Also, it cannot be excluded that the decrease in late stage germ cells due to enhanced apoptosis created an open space for early stage germ cells to accumulate. These resulted in the accumulation of DDX4 positive cells near the adluminal compartment.

Reviewer #2 (Remarks to the Author):

Comments for revised manuscript of “LCRMP-1 consummates spermiation in male reproduction by stabilizing F-actin organization via PI3K-Akt pathway balancing”.

Very little is known about the molecular mechanism of spermatogenesis. Chang J-H et al. revealed that long-form collapsing response mediator protein-1 (LCRMP-1) is involved in the spermatogenesis based on their findings including abundant expression of LCRMP-1 in testis and reduced fertility of LCRMP1 knockout (KO) mice. They found that LCRMP-1 is localized in spermiation machinery at the late stages of spermatogenesis. Using the LCRMP-1 KO mice, they showed that the absence of LCRMP-1 brings disorganized F-actin assembly in the late spermatogenesis, spermiation failure and germ-cell apoptosis, and the upregulation of Akt-mTOR signaling. In this revised manuscript, they discussed possible molecular mechanisms connecting LCRMP-1 and spermiation through Akt-mTOR pathway. The authors have earnestly addressed all of the comments raised by reviewers #1 and #2. I am satisfied with their revisions and find these results very interesting. This study becomes as a significant contribution.

1. Minor point

Fig.5c, *Lcrmp1* ^{-/-} (left side label) is missing.

Response:

We thank the reviewer’s careful reminding. We apologized this mistake and had revised it (Revised Fig. 5).

Revised Fig. 5

Fig. 5 LCRMP-1 is associated with spermiation and exhibits a stage-dependent pattern in the seminiferous tubule stages.

a Statistical analysis of the spermatogenesis stage. Testis sections were classified into 12 stages according to the histological morphology of cross sections of the seminiferous tubules. The 12 stages were further divided into five phases: I–III, IV–VI, VII–VIII, IX–X, and XI–XII. The percentages were calculated by dividing the number of seminiferous tubules in each phase by the total number of tubules analyzed (random fields, *Lcrmp-1*^{-/-}, n = 5, 87 tubules; *Lcrmp-1*^{+/+}, n = 5, 88 tubules). **b** Representative images of each phase. Sections of seminiferous tubules were prepared for H&E staining followed by morphological observation. The arrowhead in stages VII–VIII indicate residual bodies and excess cytoplasm, while the dotted circles in stages IX–XII indicate mature spermatid heads. Scale bars, 20 μm. **c** LCRMP-1 expression pattern during spermatogenesis in cycles of the seminiferous epithelium. Immunohistochemical staining of testis sections was done using eGFP antibodies to reveal the pattern of LCRMP-1 expression. The pattern in each stage was further analyzed. LCRMP-1 expression was cyclical, stage-specific, changing according to the stage of the seminiferous tubules. Black arrowheads in the white frame showing a magnified section indicate LCRMP-1 expression colocalized with cytoskeleton-like structures along the dorsal curvature of the spermatid head. Scale bars, 20 μm. **d** The histopathology of stage VII–VIII seminiferous tubules in *Lcrmp-1*^{+/+} and *Lcrmp-1*^{-/-} mice. The black arrowhead in (k) indicates a large cell with a pyknotic nucleus and reddish cytoplasm in the spermatid cell layer. Black arrowheads in (m), (n), (o), and (r) indicate spermatids with excess cytoplasm. Black arrowheads in (q), (r), and (s) indicate spermiation failure or delayed spermiation. The black arrowhead in (t) indicates degenerated spermatids or atypical residual bodies (n = 5 for *Lcrmp-1*^{+/+} and *Lcrmp-1*^{-/-} mice, 22 tubule cross sections showing stages VII–VIII). Scale bar, 20 μm.

References

- 1 Kim, J. Y., Jung, H. J. & Yoon, M. J. VASA (DDX4) is a Putative Marker for Spermatogonia, Spermatocytes and Round Spermatids in Stallions. *Reprod Domest Anim* **50**, 1032-1038, doi:10.1111/rda.12632 (2015).

- 2 Isoler-Alcaraz, J., Fernandez-Perez, D., Larriba, E. & Del Mazo, J. Cellular and molecular characterization of gametogenic progression in ex vivo cultured prepuberal mouse testes. *Reprod Biol Endocrinol* **15**, 85, doi:10.1186/s12958-017-0305-y (2017).
- 3 Kishi, K. *et al.* Spermatogonial deubiquitinase USP9X is essential for proper spermatogenesis in mice. *Reproduction* **154**, 135-143, doi:10.1530/REP-17-0184 (2017).
- 4 Endo, T., Freinkman, E., de Rooij, D. G. & Page, D. C. Periodic production of retinoic acid by meiotic and somatic cells coordinates four transitions in mouse spermatogenesis. *Proc Natl Acad Sci U S A* **114**, E10132-E10141, doi:10.1073/pnas.1710837114 (2017).
- 5 Hess, R. A. & Renato de Franca, L. Spermatogenesis and cycle of the seminiferous epithelium. *Adv Exp Med Biol* **636**, 1-15, doi:10.1007/978-0-387-09597-4_1 (2008).
- 6 Okano, T. *et al.* Classification of the spermatogenic cycle, seasonal changes of seminiferous tubule morphology and estimation of the breeding season of the large Japanese field mouse (*Apodemus speciosus*) in Toyama and Aomori prefectures, Japan. *J Vet Med Sci* **77**, 799-807, doi:10.1292/jvms.14-0411 (2015).

REVIEWERS' COMMENTS:

Reviewer #1 (Remarks to the Author):

Authors responded to all my comments and I satisfied to them.